# MMS observations of energetic oxygen ions at the duskside magnetopause during intense substorms

**Chen Zeng[1,2], Suping Duan[1], Chi Wang[1,2], Lei Dai[1], Stephen Fuselier[3,4], James Burch[3], Roy Torbert[5], Barbara Giles[6], Christopher Russell[7]**

[1]State Key Laboratory of Space Weather, National Space Science Center, Chinese Academy of Sciences, China.

[2]University of Chinese Academy of Sciences, China.

[3]Southwest Research Institute, San Antonio, Texas, USA.

[4]The University of Texas at San Antonio, San Antonio, Texas, USA.

[5]University of New Hampshire, Space Science Center, Durham, New Hampshire, USA.

[6]NASA, Goddard Space Flight Center, Greenbelt, MD, USA.

[7]University of California Los Angeles, IGPP/EPSS, Los Angeles, California, USA.

*Corresponding author:* Chi Wang (cw@spaceweather.ac.cn), Suping Duan(spduan@nssc.ac.cn)

**Abstract**

Energetic oxygen ions (1-40 keV) observed by the Magnetospheric Multiscale (MMS) satellites at the duskside magnetopause boundary layer during phase 1 are investigated. There are 57 duskside magnetopause crossing events during intense substorms (AE>500 nT) are identified. These 57 events of energetic $O^+$ at the duskside magnetopause include 26 events during the expansion phase and 31 events during the recovery phase of intense substorms. It is found that the $O^+$ density in the duskside magnetopause boundary layer during the recovery phase (0.081 $cm^{-3}$) is larger than that during the expansion phase (0.069 $cm^{-3}$). The 26 events of energetic $O^+$ ion at the duskside magnetopause during intense substorm expansion phase are all under the southward interplanetary magnetic field (IMF). There are only 7 events under northward IMF and they all occurred during the intense substorm recovery phase. The density of energetic $O^+$ at the duskside magnetopause ranges from 0.007 to 0.599 $cm^{-3}$. The maximum density of $O^+$ occurred during the intense substorm recovery phase and under southward IMF. When the IMF is southward, the $O^+$ density shows an exponential increase with the IMF $B_z$ absolute value. Meanwhile, The $O^+/H^+$ density ratio shows an exponential growth with the IMF $B_y$. These results

agree with previous studies in the near-Earth magnetosphere during intense substorm. It is suggested that
$O^+$ abundance in the duskside magnetopause boundary layer has a close relation with $O^+$ variations in the
near-Earth magnetosphere during intense substorms.
**1 Introduction**
Single charged oxygen ions ($O^+$) in the magnetosphere are exclusively from the ionosphere. They
are an important element in the mass and energy transport in the magnetospheric dynamic process,
especially during the expansion phase and recovery phase of intense substorms (e.g., Daglis et al., 1991,
1996; Duan et al., 2017; Fok et al., 2006; Ohtani et al., 2011; Ono et al., 2009; Nosé et al., 2000; Yau et
al.,1997, 2012; Kronberg et al., 2014). Processes in the magnetotail due to substorm can result in auroral
electrojet activity. This activity is generally caused by field-aligned currents increase and reflected by
the AE index (Tang and Wang, 2018). Previous studies have found that the density and energy density
of $O^+$ significantly increase with AE index in the near-Earth magnetosphere during the intense substorm
(e.g., Lennartsson and Shelley, 1986; Daglis et al., 1991, 1994; Duan et al., 2017). Lennartsson and
Shelley (1986) proposed that oxygen ions with energies less than 17 keV/e could provide 50% of the
density in the plasma sheet during disturbed geomagnetic activity. They found the increase in the $O^+$
energy density was strongest around local midnight where $O^+$ became the most abundant ion at AE~1000
nT. In the near-Earth plasma sheet (NEPS), The $O^+$ energy density has an explosively increases with AE
index in the range of larger than 500 nT during the intense substorm expansion phase (Daglis et al., 1994).
Previous studies reported that the $O^+$ from the nightside auroral region could rapidly feed in the near-
Earth magnetosphere during substorm expansion phase (e.g., Daglis and Axford, 1996; Duan et al., 2017;
Yu et al., 2013). Otherwise, the solar wind dynamic pressure also influences the oxygen content of ion
outflow from the ionosphere. Using the Thermal Ion Dynamics Experiment (TIDE) on the Polar satellite.
Elliott et al., (2001) found both the $O^+$ density and parallel flux increased with the solar wind dynamic
pressure.
The $O^+$ outflowing from the ionosphere with low energy of eV are accelerated to about 500 eV at
the high altitude polar region (e. g., Yau and André, 1997). Then they are convected tailward into the
lobe and the plasma sheet boundary layer. After $O^+$ enter the NEPS of magnetotail, they can be energized
up to tens of keV during intense substorm dipolarizations(e.g., Birn et al., 2013; Duan et al., 2017;
Fok,et al., 2006; Nosé et al., 2000; Ono et al., 2009; Yau et al., 2012). The inductive electric field
associated with substorm dipolarization is very significant for accelerating particles in the NEPS (e.g.,
Dai et al., 2014, 2015; Duan et al., 2011, 2016; Lui et al., 1999). Duan et al. (2017) reported that the $O^+$
from the lobe or the plasma sheet boundary layer were efficiently accelerated by the kinetic Alfven
eigenmode with significant unipolar electric field and rapidly feed in the NEPS during intense substorm
dipolarizations. These energetic $O^+$ in the NEPS can be injected into the inner magnetosphere and drift
westward into the duskside outer magnetosphere (e.g., Ganushkina et al., 2005).

Oxygen ions decay from the ring current can leak into the dayside magnetopause boundary layer

(e.g., Li et al., 1993; Ebihara et al., 2011). Li et al. (1993) reported that the ring current $O^+$ with tens of
keV energy interacted with the Pc 5 waves and then lost towards the dayside magnetopause.The solar
wind dynamic pressure enhancement plays a key role in the ring current particle loss into the outer
magnetosphere. This pressure enhancement pushing the magnetopause to move inward leads to a
reduction of the scale length of the magnetic field magnitude gradient along the magnetopause. The
magnetic gradient drift speed across the magnetopause will increase. So the ring current oxygen ions
along the magnetic gradient drift path can easily enter into the outer magnetosphere (Kim et al., 2005).
Ebihara et al. (2011) proposed that the field line curvature scattering was more effective on the loss of
energetic oxygen ions with its large gyro-radius. The energetic oxygen ions with pitch angle of ~90
degrees are more prone to leak into the dayside magnetopause.

The distribution of energetic oxygen ions density at the dayside magnetopause is asymmetric and it

has a close relationship with the interplanetary magnetic field (IMF) (e.g., Bouhram et al., 2005; Phan et
al., 2004; Luo et al., 2017). Bouhram et al. (2005) pointed out that the $O^+$ density in the duskside (on
average 0.053 $cm^{-3}$) magnetopause is higher than that in the dawnside (on average 0.014 $cm^{-3}$). They
found $O^+$ was the dominant contributor to the mass density (30%) on the duskside magnetopause in
comparison to 3% in the dawnside and 4% near the noon. The dawn-dusk asymmetries of the energetic
$O^+$ (>~274 keV) distribution in three different regions (dayside magnetopause, near-Earth nightside
plasma sheet, and tail plasma sheet) are also observed by Luo et al., (2017). They found that the energetic
$O^+$ distributions were mainly influenced by the dawn-dusk IMF directions and the enhancement of ion
intensity strongly related to the location of the magnetopause reconnection.

There is ample evidence that magnetospheric ions could participate in the magnetopause

reconnection and directly escape along the reconnected open field lines (e.g., Sonnerrup et al.,1981;
Fuselier et al., 1991, 2016; Slapak et al., 2012, 2015; Wang et al., 2014; Liu et al., 2015). The energetic
$O^+$ with energies larger than 3 keV in the reconnection jets at the duskside mid-latitude magnetopause
under steady southward IMF was reported by Phan et al. (2004). Zong et al., (2001) observed $O^+$ energy
dispersion due to time-of-flight (TOF) effects at the duskside magnetopause under southward IMF and
it was assumed that $O^+$ was escaping from the ring current along the reconnected field lines during steady
reconnection. However, Fuselier et al. (1989) reported that $O^+$ from the high latitude ionosphere were
not associated with any substorm cycle. $O^+$ from the high latitude ionosphere could form the $O^+$ rich
boundary layer in the low latitude magnetopause. When $O^+$ enter the reconnection jets, the reconnection
rate is likely reduced by the mass-loading but not suppressed at the magnetopause (Fuselier et al. 2019).
At present, variations of $O^+$ abundance ($O^+/H^+$) in the duskside magnetopause boundary layer
during intense substorm (AE >500 nT) with AE index and solar wind conditions (e.g. IMF $B_y$, IMF $B_z$
and solar wind dynamic pressure) are still not clear. Previous studies of $O^+$ abundance variations during
substorms are mainly focused on the magnetotail or the near-Earth region (e.g., Duan et al., 2017; Nosé
et al., 2000; Ohtani et al., 2011). The Magnetospheric Multiscale (MMS) mission gives us an opportunity
to focus on the $O^+$ in the duskside magnetopause region. In this study, we investigate statistical features
of energetic $O^+$ at the duskside magnetopause and their relations with AE index and solar wind conditions
(e.g. IMF $B_y$, IMF $B_z$, and solar wind dynamic pressure) during the intense substorms (AE >500 nT).
**2 Instrumentation and Data**
This study used data from the Magnetospheric Multiscale (MMS) mission. This mission comprises
four identical satellites that were launched on 2015 March 13 into an elliptical 28-inclination orbit with
perigee around 1.2 $R_E$ and apogee around 12 $R_E$ (Burch et al., 2016; Fuselier, et al., 2016b). The electric
field **E** is from the electric double probe (EDP) (Ergun et al., 2016; Lindqvist et al., 2016), and magnetic
field **B** is from the Fluxgate Magnetometer (FGM) (Russell et al., 2016). The plasma data are from the
Fast Plasma Investigation (FPI) and the Hot Plasma Composition Analyzer (HPCA). The FPI provides
plasma (electrons and ions) distribution functions at 32 energies from 10 eV to 30 keV. And it has a high
time resolution of 0.03 s for electrons and 0.15 s for ions in the burst mode and 4.5 s in the fast mode
(Pollock et al., 2016). The FPI does not discriminate between different ion species. While the HPCA
provides ion composition ($H^+$, $He^{++}$, $He^+$ and $O^+$) measurements in the energy range from 1 eV/q to 40
keV/q (Young et al., 2016). Although the HPCA instrument employs radio frequency (RF) unit to
artificially reduce the proton fluxes in some areas where the proton fluxes are intense, there still exists a
low level of background that affects the $O^+$ fluxes in the magnetosheath. The majority of the $O^+$ fluxes in
the magnetosphere side of the magnetopause are at energies from 1 keV to 40 keV and that band below
1keV visible in the magnetosheath side are observations outside the RF operating range and
contamination from high proton fluxes. Due to this contamination, the $O^+$ at energies from1 keV to 40
keV in the magnetopause boundary layer are considered in our study. The $O^+$ density recalculated from
the HPCA distribution functions at this energy range by using the Space Physics Environment Data
Analysis System (SPEDAS) software package. More details about SPEDAS can be found in
Angelopoulos et al. (2019). The solar wind parameters and AE index are available from the OMNI data
in CDAweb (http://cdaweb.gsfc.nasa.gov/). The data from the MMS 4 satellite are adopted in our
investigation since the data difference from other three spacecraft is negligible. This is due to spacecraft
separation and scales of particle motion.
**3 Results**
**3.1 Detailed event on 3 October 2015**
Figure 1 presents the three components of the IMF in Geocentric Solar Magnetospheric (GSM)
coordinates, solar wind dynamic pressure, as well as AU, AL, and AE index during the time of interest
from 14:30 to 16:30 UT on 3 October 2015. During this interval, the IMF $B_x$ component is negative all
the time (Figure 1a). Its maximum value is about -1 nT at ~15:16 UT. The IMF $B_y$ component is almost
negative except at ~14:32 and ~16:23 UT. The negative IMF $B_z$ component is also observed during this
interval as shown in Figure 1c. The minimum value of the IMF $B_z$ component is about -7.1 nT at ~14:30
UT. The solar wind dynamic pressure is only at the beginning of the time interval about 2 nPa. Then, it
increases sharply at 15:00 UT and reaches its maximum value about 4.4 nPa at ~15:12 UT. These solar
wind conditions led to an intense substorm (AE > 500 nT), as Figure 1g shown. The AE index is defined
as AE=AU-AL. Generally, the substorm onset time is characterized by the AL index starting to
significantly decrease and the AE index significantly increase. The interval of the AL index decreasing
from onset to its minimum is defined as the substorm expansion phase. The interval of the AL index
increasing from the minimum to the quiet time level is regarded as the substorm recovery phase. From
Figure 1e to 1g, the substorm onset time is about 14:45 UT marked by the AL index starting to sharply
decrease and AE index increase. After the AE index significantly increases and the AL index decreases
(Figure 1f), the AL and AE indexes reach their minimum and maximum values about -750 nT and 1000
nT at ~15:20 UT, respectively. This interval from ~14:45 to ~15:20 UT is regarded as the intense
substorm expansion phase. Then, the intense substorm enters the recovery phase as the AL index
gradually increases and AE index decreases after ~15:20 UT. The two blue dashed lines indicate the time
interval of the magnetopause boundary layer crossing. According to the above description, we can
identify this magnetopause boundary layer crossing occurred during the recovery phase of intense
substorm. The identification of the magnetopause boundary layer will be described later.

Figure 2 shows the overview of the magnetopause inbound crossing from 15:00 to 16:00 UT on 03

October 2015. During the magnetopause crossing, MMS 4 satellite was located at about (6.0, 8.8, -5.1)
$R_E$ in GSM as shown in the bottom of Figure 2. From top to bottom, panels 2a and 2b show the magnetic
and electric fields in GSM from FGM and EDP, respectively. Ion and electron temperatures, plasma
density and ion velocity in GSM from FPI L2 data products are shown in Figures 2c-e. Figure 2f shows
the $H^+$ and $O^+$ densities, followed by the electron and ion omnidirectional differential energy fluxes from
FPI (Figure 2g-h). The last four panels present the differential fluxes of four individual ion species, $H^+$,
$O^+$, $He^+$ and $He^{++}$ measured by HPCA, respectively. The HPCA flux in panels 2i-l has artificial striping
every 4 energy bins due to way HPCA determines the count rate over 4 energy channels in survey mode.
It is noted that the differential fluxes (Figure 2i-l) and differential energy fluxes (Figures 2g-h) have
different units. To better identify the fluxes variations at specific energies, we choose the ion and electron
fluxes from FPI in the energy flux unit. The plasma moments (e.g. ion parallel and perpendicular
temperatures, ion and electron densities, and ion velocity) from FPI shown in Figures 2c-e are all from
MMS L2 data products. They are default moments calculated over the full FPI energy range from 10 eV
to 30 keV. Note that in the magnetosheath, $O^+$ measurements suffer from fake counts at energies below
1 keV which results from high proton fluxes contamination, as the red box in Figure 2j shown. So the
spurious counts should be excluded in the plasma moments calculation. The $O^+$ density shown in Figure
2f is recalculated from HPCA distribution functions at energies from 1 keV to 40 keV. Due to $H^+$
measurements from HPCA is accurate and the $H^+$ mean energy in the magnetosheath is typically 0.3 keV,
we adopted the default $H^+$ density from HPCA L2 data products which computed over the full HPCA
energy range from 1 eV to 40 keV, as the red line shown in Figure 2f.
The different regions encountered by MMS4 during the interval of 15:00 to 16:00 UT are marked
by the colored bar at the top of Figure 2, with the magnetosheath shown in orange, the outer
magnetosphere shown in blue, and the magnetopause boundary layer shown in green. From 15:00:00 to
15:25:10 UT, MMS4 was located in the magnetosheath. This region is characterized by the southward
magnetic field, low ion and electron temperatures (a few hundred eV for ions and tens of eV for electrons,
Figure 2c) with relatively high densities (on the order of ~20 cm$^{-3}$, Figure 2d), and stable ion flow speed
of about 100 km/s. There are also very high fluxes at energies centered around 100 eV (nominal
magnetosheath energy) for electrons (Figure 2g) and at energies centered around 1 keV for ions (Figure
2h, also see H$^+$ fluxes in Figure 2i and He$^{++}$ fluxes in Figure 2l) in the magnetosheath. While the O$^+$ and
He$^+$ fluxes above 1 keV nearly disappear in the magnetosheath (Figure 2j and 2k). From Figure 2j, the
majority of the O$^+$ fluxes at energies below 1 keV visible in the magnetosheath are the result of
contamination from the high proton fluxes, as the red box indicated.
The primary magnetopause crossing from the magnetosheath into the magnetosphere lasts about 12
min, from about 15:25:10 to 15:36:50 UT. Partial encounters of the magnetopause by MMS4 occurred
around 15:43:15, 15:47:10, 15:53:00 UT and etc. The magnetopause boundary layer is identified by the
plasma moments and the electromagnetic field. The plasma density and temperature at the magnetopause
are between the corresponding values of the magnetosphere and the magnetosheath, as shown in Figure
2d and 2c. The magnetopause boundary layer can also be identified by the significant increases in
electron fluxes at energies about several hundred eV and ion fluxes at energies around ~10 keV, as shown
in Figure 2g and 2h, respectively. During this time of interest, the B$_z$ component rotated from southward
to northward and back again several times before finally became northward when MMS 4 entered the
magnetosphere. The energetic O$^+$ density (1-40 keV) is around 0.018 cm$^{-3}$ within the magnetopause
boundary layer as shown in Figure 2f. The corresponding H$^+$ and O$^+$ fluxes at specific energies and their
densities (shown in Figure 2f) were averaged in this region.
After 15:36:50 UT, MMS4 entered the magnetosphere which is identified by the observations of
the northward magnetic field (Figure 2a), much lower plasma densities (on the order of ~1 cm$^{-3}$) with
respect to the densities in the magnetosheath (Figure 2d), higher plasma temperatures (Figure 2c, several
keV for ions and a few hundred eV for electrons), and a small bulk ion flow speed. Higher fluxes at
energies around several keV for electrons (Figure 2g) and at energies centered around ~10 keV for ions
(Figure 2h) also indicate that the MMS4 was in the magnetosphere. Finally, the presence of O$^+$ and He$^+$
at energies about ~10 keV is also used as a marker to verify that MMS4 was in the magnetosphere (Figure
2j and 2k).

**3.2 Statistical 57 events of energetic $O^+$ at the duskside magnetopause during intense substorms**

Based on the in-situ measurements of the dayside magnetopause crossings by MMS satellites in
phase 1, we identified the duskside magnetopause crossing event (complete magnetopause crossing from
the magnetosheath to the magnetosphere, vice versa) from the summary plot in
https://lasp.colorado.edu/mms/sdc/public/plots/. Then we plotted the more detailed overview figure of
these events to identify the magnetopause boundary layer, as Figure 2 shown. Only events with AE index
larger than 500 nT during the magnetopause boundary layer crossings interval were selected. There are
57 events of the dusksideside magnetopause boundary layer crossing during intense substorm satisfied
with the above criterion. In our statistical study, the mean values of the $H^+$ and $O^+$ fluxes at specific
energies and their densities are calculated in the magnetopause boundary layer. Correspondingly, the
solar wind dynamic pressure, IMF $B_y$, $B_z$ and AE index from the OMNI data system were averaged
during the magnetopause boundary layer crossing time interval, as the two blue dashed lines shown in
Figure 1. The phase of the substorm is determined from the variations of AU, AL and AE indexes, as
mentioned before. For better follow-on studies, we add more detail information about 57 energetic $O^+$
events into an appendix. From the appendix, we can easily draw the conclusion that the $O^+$ density in the
duskside magnetopause during the recovery phase (0.081 $cm^{-3}$) of intense substorm is larger than that
during the expansion phase (0.069 $cm^{-3}$).
Figure 3 displays the locations of 57 energetic $O^+$ events at the duskside magnetopause (-5.7 $R_E$ <
$Z_{GSM}$ < 1.7 $R_E$) during intense substorms projected into the $XY_{GSM}$ plane. The blue curve line represents
the nominal magnetopause, which is obtained by the magnetopause model of Shue et al., (1998) when
the IMF $B_z$ is about -3.21 nT and solar wind dynamic pressure (Psw) is ~2.87 nPa (averaged for the 57
events). The diamond and circle represent the event at the duskside magnetopause during the intense
substorm expansion phase and recovery phase, respectively. The $O^+$ density and the $O^+/H^+$ density ratio
are shown by the colored diamonds and circles at the corresponding magnetopause locations in Figures
3a and 3b, respectively. Among the 57 events of energetic $O^+$ at the duskside magnetopause during
intense substorms, there are 26 events that occurred during the expansion phase of intense substorms and
31 events occurred during the recovery phase. The maximum density of energetic $O^+$ is found during the
intense substorm recovery phase, as presented in Figure 3a.

Figure 4 presents the relationship between the energetic $O^+$ at the duskside magnetopause and AE

index during intense substorms. From top to bottom, panels show the $O^+$ and $H^+$ densities (Figure 4a),
the $O^+/H^+$ density ratio (Figure 4b) and $O^+/H^+$ particle fluxes ratios at different energy ranges (Figure 4c),
respectively. The energy channel ranges for $O^+$ and $H^+$ in the HPCA are the same. So the $O^+/H^+$ particle
fluxes ratio is directly defined as the ratio between mean values of their fluxes, respectively. The particle
fluxes are chosen at energies ~1 keV (energy range from 987.82 to 1165.21 eV), ~10 keV (energy range
from 9.97 to 11.77 keV), ~20 keV (energy range from 19.31 to 22.78 keV) and ~35 keV(energy range
from 31.69 to 37.39 keV). The error bars indicating the 90% confidence interval (CI) are also overplotted
in each point. The confidence interval is based on the following formula:
$$\bar{x} - k\frac{s}{\sqrt{n}} < \mu < \bar{x} + k\frac{s}{\sqrt{n}}$$

Where $\bar{x}$, $s$ and $n$ are the mean value, standard deviation and the sampling number of observations,

respectively. $k$ in the above formula can be determined by calculating a 90% confidence interval for each
events (the $k$ value is 1.65). Figure 4a shows that energetic $O^+$ density at the duskside magnetopause
during intense substorms is in the range from 0.007 to 0.599 $cm^{-3}$. The maximum density value of
energetic $O^+$ at the duskside magnetopause during intense substorm recovery phase is presented at the
higher AE index about 606 nT. The $O^+/H^+$ density ratio decreases with AE index from 900 to 1100 nT.
The variations of $O^+$ density and $O^+/H^+$ density ratio with AE index do not show obvious difference
between during the expansion phase and the recovery phase.

Figure 5 shows the relationship between the energetic $O^+$ at the duskside magnetopause and IMF

$B_y$ during intense substorms. The format of Figure 5 is the same as that of Figure 4. Figure 5a shows that
the $O^+$ and $H^+$ densities decrease with IMF $B_y$ from -6 to 0 nT and increase with IMF $B_y$ from 4 to 8 nT.
From Figure 5b, the $O^+/H^+$density ratio shows an exponential growth with the IMF $B_y$. Based on the
scatter plot in Figure 5b, we can define linear functional dependence between the logarithm of $O^+/H^+$
density ratio and IMF $B_y$, as Eq. (1) shown. And the corresponding correlation coefficients is 94%. The
correlation coefficient close to 100% indicates that there is a great correlation.
$$\log\frac{n_{O^+}}{n_{H^+}} = 0.126 * \text{IMF By} - 5.174 \tag{1}$$

The dependency is constructed using a linear least-squares fit unless otherwise stated. The $O^+/H^+$
particle flux ratio at energy ~10 keV, ~20 keV and ~35 keV also show an obvious exponential increase
with IMF $B_y$. This dependency is consistent with Welling et al. (2011) simulation results found in the
ring current.
Figure 6 shows the relationship between the energetic $O^+$ at the duskside magnetopause and IMF
$B_z$ during intense substorms. The format of Figure 6 is the same as that of Figure 4. Figure 6a and 6b
both present that among 57 events of energetic $O^+$ at the duskside magnetopause boundary layer during
intense substorm, there are 50 events under southward IMF and only 7 events under northward IMF. It
is noted that 26 events occurred during the expansion phase of intense substorms which are all under the
southward IMF conditions, as the blue points shown. Meanwhile, the events that occurred under
northward IMF are all during the intense substorm recovery phase, as the right red points with positive
IMF $B_z$ shown. From -10 to 0 nT, the $O^+$ density shows an obvious decrease with IMF $B_z$. To better
describe this variation trend, the empirical functional relation between the logarithm of $O^+$ density and
IMF $B_z$ (from -10 to 0 nT) is established in Eq.(2) and the corresponding correlation coefficient is 94%.
While the $O^+$ density has a positive correlation with IMF $B_z$ from 0 to 5 nT.
$$\log n_{O^+} = -0.163 * \text{IMF Bz} - 3.737 \qquad (2)$$

From Figure 6b, the $O^+/H^+$ density ratio during the recovery phase decrease with IMF $B_z$ from about -2
to 2 nT. The maximum density of energetic $O^+$ at the duskside magnetopause is under southward IMF.
Meanwhile, the maximum $O^+/H^+$ density ratio at the duskside magnetopause is also under southward
IMF.
Figure 7 displays the relationship between the energetic $O^+$ at the duskside magnetopause and solar
wind dynamic pressure during intense substorms. The format of Figure 7 is the same as that of Figure 4.
Figure 7a presents that the $O^+$ density at the duskside magnetopause during intense substorms has a
positive correlation with the solar wind dynamic pressure. The empirical functional relation between the
logarithm of $O^+$ density and solar wind dynamic pressure (from 1 to 4.5 nPa) is also established in Eq.(3)
and the corresponding correlation coefficient is 94%.
$$\log n_{O^+} = 0.325 * \text{Psw} - 4.061 \qquad (3)$$

From Figure 7b, the $O^+/H^+$ density ratio during recovery phase show a decrease from about 2.5 to 3 nPa.
It is noted that the $O^+/H^+$ density ratio increase with solar wind dynamic pressure from about 3 to 4 nPa.
The maximum density of energetic $O^+$ at the duskside magnetopause, ~0.599 $cm^{-3}$ take place at solar
wind dynamic pressure is about 3.9 nPa. While the maximum $O^+/H^+$ density ratio at the duskside
magnetopause appeared at solar wind dynamic pressure around 2.2 nPa. More details can be found in the
appendix.
**4 Discussion**
Energetic $O^+$ (1-40 keV) with high density are observed by MMS satellites at the duskside
magnetopause during the expansion phases and recovery phases of intense substorms, as displayed in
Figure 3a. The density of energetic $O^+$ is in range from 0.007 $cm^{-3}$ to 0.599 $cm^{-3}$ at the duskside
magnetopause boundary layer during intense substorms. In a companion paper from Zeng et al. (2019),
they study the $O^+$ abundance variations on the solar wind conditions at the dayside magnetopause
boundary layer and not specific to the events that occurred during intense substorm. The mean value of
the $O^+$ density at the duskside magnetopause boundary layer is 0.038 $cm^{-3}$ in that paper. While during the
intense substorm, the $O^+$ density increase to 0.075 $cm^{-3}$ in this study. There are two reasons for this high
density of energetic $O^+$ observed during the intense substorm. The first is the time interval for the
observations. Our observations are during intense substorms expansion phase and recovery phase. Daglis
et al. (1991) proposed that energetic $O^+$ were significantly higher in the NEPS in the magnetotail after
intense substorms onset. The impulsive electric field accompanied by intense substorm dipolarization
plays a key role in the energization and sunward transfer of oxygen ions in the duskside of midnight
plasma sheet in the magnetotail (e.g., Fok et al., 2006; Nosé et al., 2000). These energetic $O^+$ (tens of
keV) can be transported sunward into the duskside magnetopause boundary layer. The second reason for
the high densities is the locations of the observations. Our observations are near the duskside
magnetopause. This region is easily accessible by energetic $O^+$ during intense geomagnetic activity
(Fuselier et al. 2016a). Phan et al. (2004) pointed out that energetic $O^+$ with very high density 0.2-0.3
$cm^{-3}$ in the reconnection jets at the duskside mid-latitude magnetopause were observed by Cluster.
During dynamic periods and intense substorms time, light ions yielded more symmetric patterns in
density than heavy ions and the $O^+$ patterns in the active plasma sheet are a function of IMF conditions
(Winglee and Harnett 2011. Winglee et al. 2005). Welling et al. (2011) used multispecies MHD and the
PWOM to drive a ring current model and found that positive IMF $B_y$ pushed the stronger $O^+$
concentrations toward the duskside at a geocentric distance of about 6.6 $R_E$. This $O^+$ density duskward
preference with positive IMF $B_y$ in the NEPS is similar to our results. It may indicate that $O^+$ in the
magnetopause boundary layer enhancing with IMF $B_y$ is due to the local time variations of $O^+$ in the
NEPS tied to IMF $B_y$. Our result of $O^+$ density increase with IMF $B_y$ also agree with Kronberg et al.,
(2012). They showed for 10 keV $O^+$ strong increasing under the duskward IMF indicated by the clock
angle in the inner magnetosphere. It is suggested that the $O^+$ abundance at the duskside magnetopause
has a corresponding relation with the $O^+$ in the duskside near-Earth magnetosphere during intense
substorm. The $O^+$ path from the cusp to the magnetotail is asymmetric and it has the best correlation with
the IMF directions. This path asymmetry mainly controlled by the IMF $B_y$ may influence the $O^+$
abundance at the duskside magnetopause. When the IMF $B_y$ is positive, the $O^+$ from northern/southern
cusp tends to flow toward the dawnside/duskside. The transport path for negative IMF $B_y$ is more
symmetric but shows some evidence for a reversed asymmetry when the negative IMF $B_y$ is large enough.
While the IMF $B_z$ has little influence on the asymmetry (Liao et al., 2010).

Due to not enough events occurred under northward IMF were observed, the influence of IMF $B_z$

on the $O^+$ abundance (1-40 keV) during intense substorm is not clear. While Luo et al. (2017) found that
the $O^+$ intensity (> ~274 keV) was significantly higher under southward IMF than that under northward
IMF, especially at the duskside magnetopause. Zeng et al. (2019) also showed that the duskside
asymmetry of $O^+$ density (1-40 keV) in the dayside magnetopause under northward IMF was less obvious
than under southward IMF when the IMF $B_y$ was the same. Under the southward IMF, the interactions
between the solar wind and the magnetosphere become active. The inductive electric field or magnetic
field gradient related to magnetic reconfiguration will enhance with negative IMF $B_z$. So the large scale
dawn-dusk electric field drift along with the gradient-curvature drift can force oxygen ions convect to
the duskside magnetopause boundary layer (Kronberg et al., 2015; Luo et al., 2017).

In this statistical study, there are 50 magnetopause boundary layer crossing events during intense

substorm under southward IMF with respect to 7 events under northward IMF. Choosing the intense
substorm may increase the probability of observing the events under southward IMF quite significantly.
Among 57 events of energetic $O^+$ near the duskside magnetopause, there are 26 events during intense
substorm expansion phase which are all under the southward IMF, as the blue circle shown in Figure
6b.There are only 7 events under northward IMF in our study and they all occurred during the intense
substorm recovery phase. But what relation between the IMF directions and phase of substorm is out of
scope for this article.
Previous studies demonstrated that the oxygen ions that originate from the aurora region could
rapidly feed in the NEPS during intense substorms expansion phase (e.g., Daglis and Axford, 1996; Duan
et al., 2017; Yu et al., 2013). Oxygen ions can be efficiently energized in the NEPS during intense
substorm dipolarization (e.g., Duan et al., 2017; Fok et al., 2006; Nosé et al., 2000). Under southward
IMF conditions, these energetic oxygen ions in the NEPS can be convected sunward and drift westward.
As a result, the energetic $O^+$ arrived near the duskside magnetopause can participate in the magnetopause
reconnection and escape along reconnected field lines during intense substorm expansion phase, as
reported by Wang et al. (2014) and Zong et al. (2001). When $O^+$ participate in the reconnection jets, the
reconnection rate will likely be reduced by the mass-loading but not suppressed at the magnetopause
(Fuselier et al. 2019). Whether these energetic $O^+$ at the duskside boundary layer could suppress the
intense substorm need further investigation.
**5 Summary and conclusions**
Using the measurements from MMS satellite during the phase 1, we have studied 57 events of the
energetic $O^+$ (1-40 keV) at the duskside magnetopause boundary layer and their variations on the solar
wind conditions (IMF $B_y$, IMF $B_z$ and solar wind dynamic pressure) during intense substorm expansion
phases and recovery phases. According to the above analysis, we can draw our main conclusions as
follows. In our 57 events of energetic $O^+$ at the duskside magnetopause boundary layer, there are 26
events during the expansion phase of intense substorms and 31 events during the recovery phase. It is
noted that the mean values of the $O^+$ density during the expansion phase and recovery phases are 0.069
$cm^{-3}$ and 0.081 $cm^{-3}$, respectively. And the maximum $O^+/H^+$ density ratio occurred during the intense
substorm recovery phase. It is found that 26 events of energetic $O^+$ at the duskside magnetopause during
intense substorms expansion phase are all under the southward IMF conditions, and only 7 events under
northward IMF which are all during the intense substorm recovery phase. The $O^+$ density shows an
exponential increase with IMF $B_z$ absolute value under the southward IMF. Similarly, it also presents an
exponential growth with solar wind dynamic pressure, and the empirical functional relations are
established. Like previous studies during substorm in the near-Earth magnetosphere, The $O^+/H^+$ density
ratio in the duskside magnetopause boundary layer enhance with the IMF $B_y$. It is suggested that the $O^+$
abundance in the duskside magnetopause boundary layer has a close correlation with the $O^+$ variations
in the near-Earth magnetosphere during intense substorm.
**Data availability**
All data used in this study are publicly accessible. MMS data are available at the MMS Science
Data Center (https://lasp.colorado.edu/mms/sdc/public/). The OMNI data can be downloaded from the
NASA    Goddard    Space    Flight    Center    Coordinated    Data    Analysis    Web
(CDAWeb:http://cdaweb.gsfc.nasa.gov/).
**Competing interests**
The authors declare that they have no conflict of interest.
**Author contribution**
C. Z. conducted the majority of the data processing, analysis and writing for this study. S.P.D, C.W,
L.D and S.F participated in the interpretation of the data and modified this paper. J.B, R.T, B.G and C.R
produced the data and controlled the data quality. All the authors discussed the results and commented
on the paper.
**Acknowledgments**
We acknowledge the entire MMS team for providing high-quality data. This work is supported by
the National Natural Science Foundation of China grants 41874196, 41674167; 41731070, 41574161
and 41574159; the Strategic Pioneer Program on Space Science, Chinese Academy of Sciences, grants
XDA15052500, XDA15350201 and XDA15011401; the NSSC Research Fund for Key Development
Directions and in part by the Specialized Research Fund for State Key Laboratories.

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

 **Figure Captions**

621

**Figure 1.** The three components of the IMF Bx, By, Bz in the Geocentric Solar Magnetospheric

coordinates, solar wind dynamic pressure, as well as AU, AL and AE index from CDAweb OMNI data.

The two blue dashed lines indicate the interval of the magnetopause boundary layer crossing.

**Figure 2.** The energetic $O^+$ is observed at the magnetopause during an intense substorm on 03 October

2015 by MMS 4. From top to bottom are (a) the magnetic field three components, Bx (blue line), By

(gree line), Bz (red line) and the total magnitude Bt (black line), (b) the electric field three components,

Ex (blue), Ey (gree) and Ez (red), (c) ion parallel (red) and perpendicular (black) temperatures, as well

as electron parallel (blue) and perpendicular (green) temperatures, (d) The density of ion (green) and electron (blue), (e) three components of the ion velocity, (f) the H$^+$ (over the full HPCA energy range from 1 eV to 40 keV) and O$^+$ (at energies from 1 keV to 40 keV) densities, (g-h) electron and ion omnidirectional differential energy fluxes (keV/(cm$^2$ s sr KeV)$^{-1}$), (i) to (l) present differential particle fluxes (cm$^2$ s sr eV)$^{-1}$ of H$^+$, O$^+$, He$^+$, He$^{++}$, respectively. The Geocentric Solar Magnetospheric coordinate system is adopted. The thick bars at the top of the panel present different regions encountered on this magnetopause crossing event. The orange and blue bars represent the magnetosheath and the magnetosphere, respectively. The green bar represents the magnetopause boundary layer. The black horizontal line in figure 2j is at 1 keV and the O$^+$ contamination from high H$^+$ fluxes is indicated by the red box. The FPI data in Figure 2c-e and g-h are from FPI L2 data products and in the fast mode.

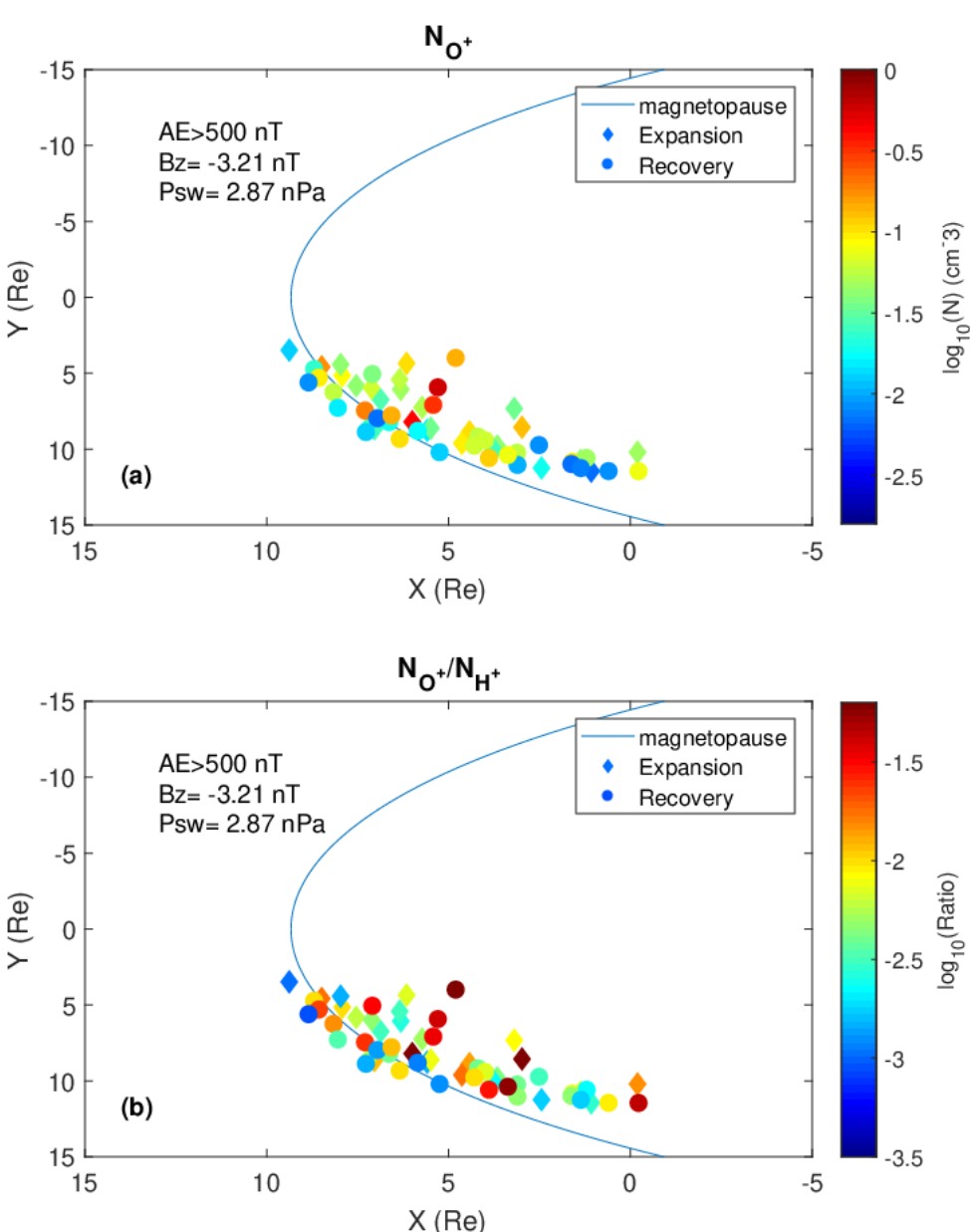

640

Figure 3. Maps of 57 events of energetic $O^+$ at the duskside magnetopause during intense substorms with
AE index larger than 500 nT in $XY_{GSM}$ plane. The $O^+$ density and the density ratio of $O^+/H^+$ are shown
by the color signatures at the corresponding magnetopause location in Figure 3a and 3b, respectively.
The blue curve line represents the nominal magnetopause. The diamond and circle represent the event at
the magnetopause during the intense substorm expansion phase and recovery phase, respectively.


Figure 4. The relationship between the energetic $O^+$ at the duskside magnetopause and AE index during
intense substorms. From top to bottom, panels show the $O^+$ and $H^+$ densities (Figure 4a), the $O^+/H^+$

density ratio (Figure 4b), and the O⁺/H⁺ particle flux ratio (Figure 4c), respectively. Error bars indicate
90% confidence intervals.

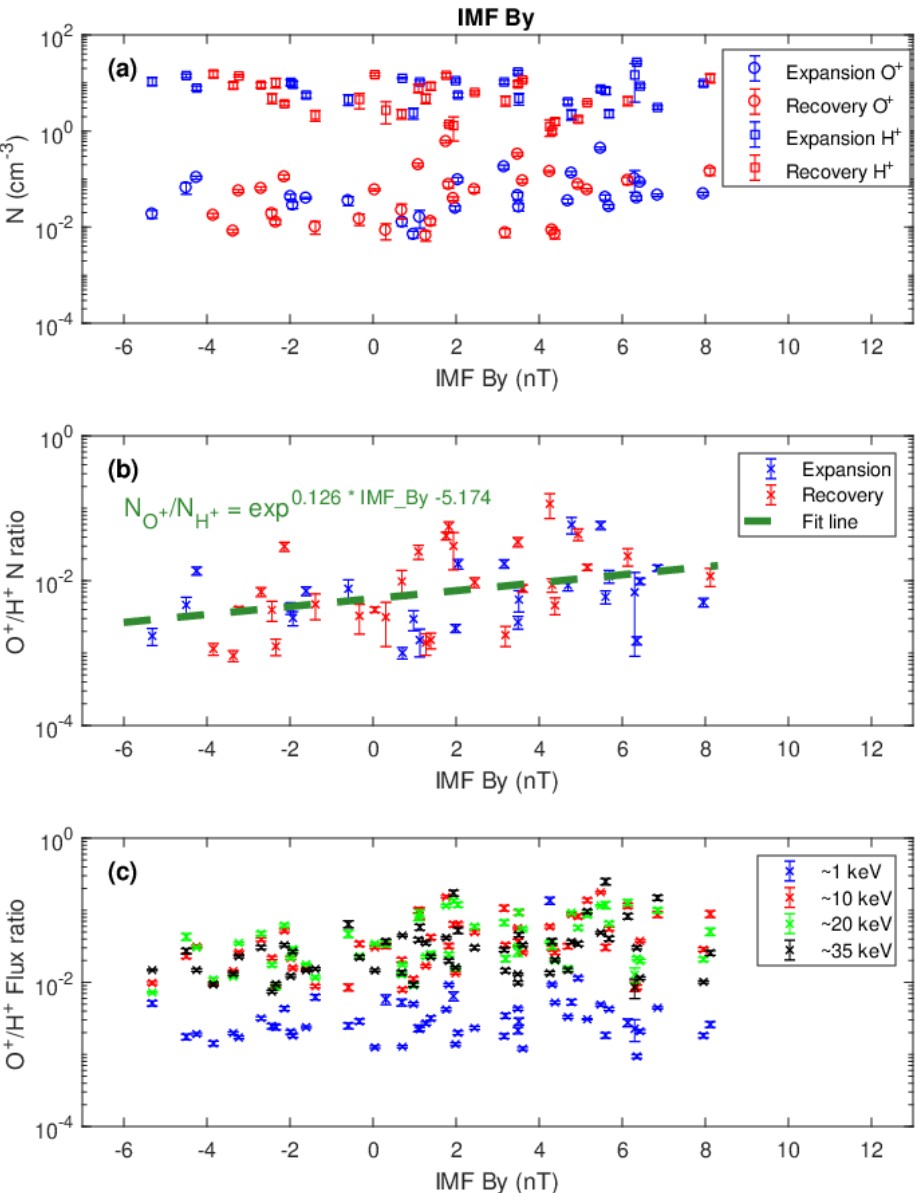


**Figure 5.** The relationship between the energetic $O^+$ at the duskside magnetopause and IMF By during
intense substorms. The format is the same as that of Figure 4.

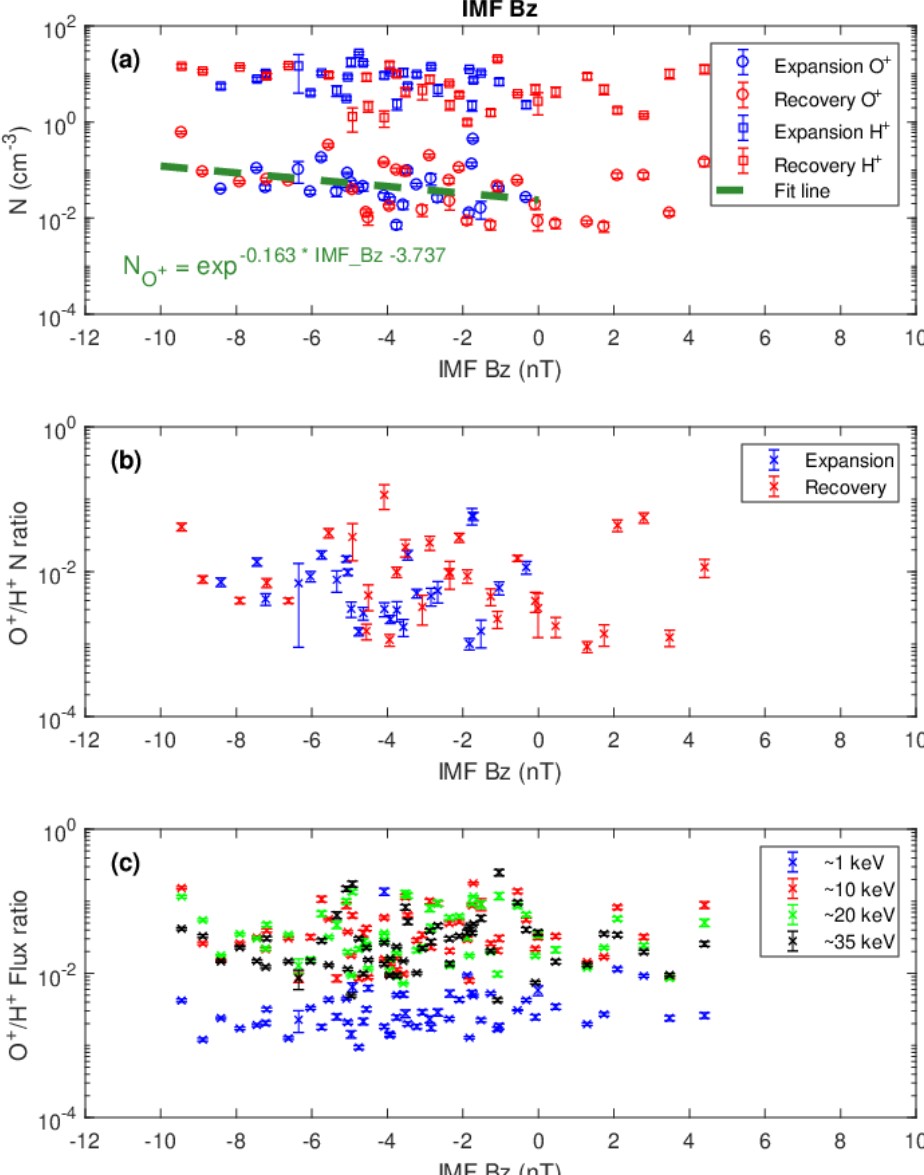


**Figure 6.** The relationship between the energetic O[+] at the duskside magnetopause and IMF Bz during
intense substorms. The format is the same as that of Figure 4.

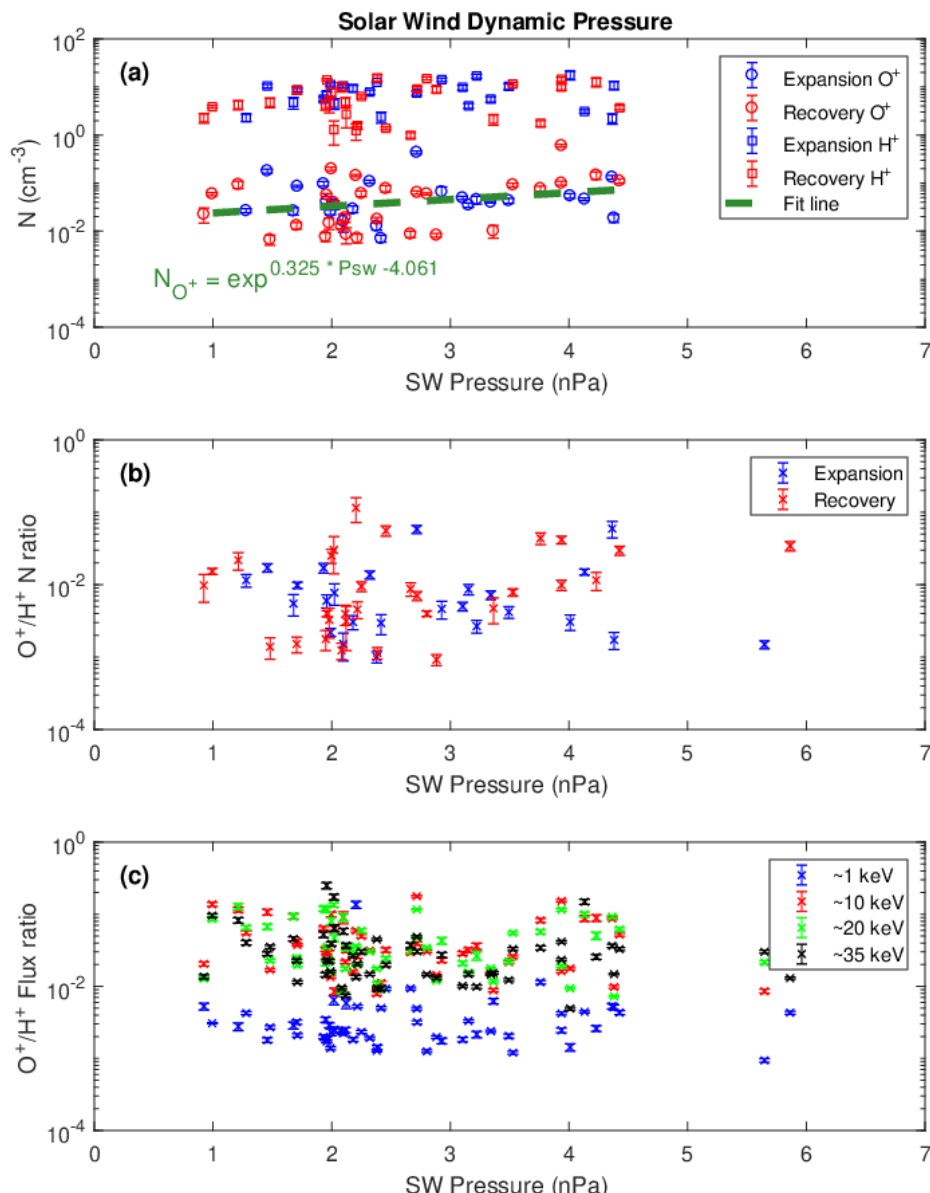


**Figure 7.** The relationship between the energetic O[+] at the duskside magnetopause and solar wind
dynamic pressure during intense substorms. The format is the same as that of Figure 4.