# Peer review of "MMS observations of energetic oxygen ions at the duskside magnetopause during intense substorms"

_Annales Geophysicae, 2019_

## Referee Comment (RC1) · Anonymous Referee #1 · 13 Aug 2019

This manuscript describes statistical analysis of energetic oxygen ion properties at the low-latitude noon-dusk magnetopause. Although the results are of interest to the current space science community, more descriptions and detail are needed to clarify the results and the methods used to derive them. This manuscript may be acceptable for publication in Annales Geophysicae following careful consideration of and adequate responses to the comments given below.

Line 138: The authors should clarify the information on which instrument dataset was used for each data product. Were the moments shown in Figure 2c-2e recalculated from the FPI distribution functions? Or are they the default moments calculated over

the full FPI energy range?

Line 140 (Figure 2f): The calculations performed to derive the >1keV O+ density need to be described to inform the reader how the HPCA energy ranges were specified for those calculations. If a software package was used, then details of the software package and a citation to it should be included. The >1keV H+ density could also be plotted in this figure panel.

Line 158-164: The magnetopause identification criteria are not very convincing. Recommend carefully defining these criteria, as all statistics are derived based on the magnetopause identification. Recommend the authors review identification criteria used in previous works. For example, Haaland et al. (2016) and (2019) describe magnetopause observations by Cluster and THEMIS:

Haaland, S., Reistad, J., Tenfjord, P., Gjerloev, J., Maes, L., DeKeyser, J., Maggiolo, R., Anekallu, C., and Dorville, N. (2014), Characteristics of the flank magnetopause: Cluster observations, J. Geophys. Res. Space Physics, 119, 9019–9037, doi:10.1002/2014JA020539.

Haaland, S., Runov, A., Artemyev, A., & Angelopoulos, V. (2019). Characteristics of the flank magnetopause: THEMIS observations. Journal of Geophysical Research: Space Physics, 124, 3421–3435. https://doi.org/10.1029/2019JA026459.

Paschmann et al. (2018) describe magnetopause identification and observations by MMS:

Paschmann, G., Haaland, S. E., Phan, T. D., Sonnerup, B. U. Ö., Burch, J. L., Torbert, R. B., et al. (2018). Large-scale survey of the structure of the dayside magnetopause by MMS. Journal of Geophysical Research: Space Physics, 123, 2018–2033. https://doi.org/10.1002/2017JA025121.

Line 180: More details are needed to describe how the mean values of the H+ and O+ fluxes and densities were calculated.

Line 184: A more detailed description of how substorm phase (i.e. expansion phase or recovery phase) was defined based on AE index is needed. The authors should use Figure 1 AE index to aid in their description.

Line 202-209: Several narrow energy ranges used for comparing the O+/H+ density ratio are noted. It is important to describe for the reader how these energy ranges were used in the density ratio calculations. In addition, a description of why these energy ranges were chosen should be included. Did the authors consider calculating the density ratio for all energies >1 keV instead of calculating the ratios over individual energy ranges? A comparison of density ratios using both methods may be helpful to decide which method to use. Such procedural information on which analysis methodology was chosen could be included in an appendix.

Line 218-248: Figures 5, 6, 7 all show comparisons of the O+/H+ density ratio. After addressing the previous comment on Line 202-209 on why separate narrow energy ranges were chosen instead of using a broad energy range, the authors may need to revise panel (b) of these three figures. For example on Line 240: Are the O+ and H+ densities referred to in this section calculated from one of the energy ranges discussed in Line 202-209? Greater detail and explanation are needed.

Line 254-256: After addressing the above comments on how the ion densities were calculated, the authors should briefly address whether these comparisons of density across missions are relevant. For example, if the O+ density (calculated over defined HPCA energy range) is higher than seen by Cluster (calculated in what energy range and using which instrument?), what does this mean? Were the instrument energy ranges equivalent or similar? Otherwise, the direct comparison may not be meaningful.

Line 305: Since 31 events is not a large number, recommend the authors produce a table listing the dates and times of each of these events so that others in the space science community can also investigate the events for follow-on studies. Such a table could go in an appendix.

All the references in the manuscript need to be checked. For example, all the MMS instrument papers were referenced but do not appear in the references list. It is likely many other references have been missed.

Line 106: Pollock et al. (2016) is referenced but does not appear in the references list

Line 105: Russell et al. (2016) is referenced but does not appear in the references list

Line 104: Ergun et al. (2016) is referenced but does not appear in the references list

Line 104: Lindqvist et al. (2016) is referenced but does not appear in the references list

Line 107: Young et al. (2016) is referenced but does not appear in the references list

---

## Referee Comment (RC2) · Anonymous Referee #2 · 21 Aug 2019

The manuscript by Zeng et al., on "MMS observations of energetic oxygen ions at the low-latitude duskside magnetopause during intense substorms" shows that energetic oxygen abundance in the magnetopause is regulated by the IMF By direction, although IMF Bz plays a minor role. The manuscript is mainly well written and organized. However, the conclusions are not well supported by the observations on my opinion. I think the manuscript has potential to be published in Annales Geophysicae after considering the following comments and suggestions:

Major comments:

1. One of the conclusions of the manuscript is that particles are transported from the

tail towards the day side. To make such a conclusion more rigid one should show the anisotropy of the particle distributions, which would indicated that particles move from the tail towards the dayside. The oxygen ions could also come from other sources such as inner magnetosphere (filled directly from the nightside aurora into the ring current), from the diamagnetic cavities/cusp (e.g. Slapak et al., Ann. Geophys. 2013, 10.5194/angeo-31-1005-2013).

2. I am not sure if one could make firm conclusions about dependence on the IMF Bz, if from 31 events only 6 events were observed during northward IMF. On my opinion the statistics is too poor for that.

3. The "intense substorms" are discussed in this study. Were these substorms associated with magnetic storms? Or these are pure substorm events? What is the reason for choosing intense substorms? Inlcuding other substorms may increase the statistics on the IMF dependence.

4. Introduction, first two paragraphs can be merged as they contain repeating information about acceleration during dipolarizations. The second paragraph is not completely logical. It would make more sense to describe acceleration of O+ starting from the polar regions, then lobe, dipolarizations and then discuss drift. The sentence in lines 43-45 discussing acceleration of electrons during dipolarizations is not really needed as there is a number of references about acceleration of oxygen during dipolarizations in lines 51-53 and the whole text is about O+.

5. lines 90-91, "At present, O+ near the dayside low-latitude magnetopause during substorm expansion phase and recovery phase are still not understood" –> What exactly do you mean under not understood? Which scientific questions are still open? Which questions do you try to answer?

6. lines 91-93, there is paper by Luo et al., JGR, 2017, 10.1002/2016JA023471, in which the energization of O+ at the dayside is discussed. The study also discusses asymmetries of the energetic oxygen due to IMF By and Bz directions. Both IMF By

and Bz influence the oxygen abundance at higher energies. However, this is large statistical study and not only cases for the intense substorms. This can be discussed.

7. lines 125-126, 130-131, 180-181, please provide more precise definition of the substorm onset and recovery phase. For example in paper by Newell and Gjerloev, JGR, 2011, 10.1029/2011JA016779, is a nice example on how to define substorm onset, also using more precise SML index available at the SuperMAG. I do not think that definition when "AE index significantly increases" is a precise one. I do not think that one should provide twice the information about substorm onset in lines 125-126 and 130-131. I would remove the second sentence.

8. lines 179-180, actual observations of the IMF and solar wind dynamic pressure could be used directly from the MMS observations at the magnetopause crossings. This would be much more precise.

9. lines 277-278, For higher energies the larger statistics one can clearly see that the stronger duskward asymmetry in the plasma sheet and the dayside magnetosphere is observed under the southward IMF, e.g. Luo et al., JGR, 2017. One should mention that no influence of IMF Bz is observed in case of the energies below 40 keV and for 31 intense substorm events.

10. lines 286-287, the energetic O+ occurs predominantly under southward IMF. Here I would say that it was chosen to be like this. Choosing the intense substorms one increases the probability of observing the southward IMF quite significantly. This also contradicts to statement in the lines 277-278, that IMF Bz does not influence abundance of O+ at the magnetopause. There is not enough provided data to conclude so. By increasing the number of events under the northward IMF one may see a different picture. One can see pretty nice trend in Figure 6b, that the abundance is increasing with the decrease of IMF Bz at least for the expansion phase. Generally on my opinion there is not enough statistics in this study to make conclusions about IMF dependence. One should expand the statistics.

11. lines 304-306, This conclusion is not supported by the observations. Just looking at the scatter points of the number density, I do not see a statistically significant difference between these two phases. One should either show fits to those points or bin them according to some parameters and show that the difference is significant.

12. lines 313-315, energetic oxygen ions also indicate the transport at the dayside magnetosphere (e.g. Liao et. al, JGR, 2010, 10.1029/2010JA015613). These different transports is hard to distinguish (e.g. Luo et al., JGR, 2017).

13. Figures 4-7, just looking at the scatter plots it is hard to make certain conclusions. One should either bin the points to show average trend or fit them with some dependences and increase the number of events.

Minor comments:

1. line 19, What is the energy range of the oxygen observations used in this study? Please indicate the upper energy limit in the abstract. This is important to know when assessing the number densities.

2. line 45, I did not find the reference to Lui et al., 1999 in the reference list.

3. line 47, "during activity geomagnetic disturbance" –> "during disturbed geomagnetic activity"

4. line 55, "[e.g. Yau and Andre, 1997]. And then..." –> "[e.g. Yau and Andre, 1997]. Then..."

5. line 85, please remove one "However".

6. lines 106-107, Does HPCA distinguish between O+, N+ and C+? Or what measured is actually the CNO group?

7. line 122, "At the beginning of the time interval, the solar wind dynamic pressure..." –> The dynamic pressure is only at the begging of the time interval about 2 nPa.

8. lines 124-125, I would change to " These solar wind conditions led to the intense substorm (AE>500 nT).

9. lines 148-150, Figure 2, I would say that the fluxes at energies below 2 keV in Figure 2j is also contamination. This should be mentioned also in Figure caption and even better when it is indicated on plot itself.

10. line 195, I would remove "On the other hand".

11. lines 221-224, These results also agree with Kronberg et al., JGR, 2012, 10.1029/2012JA018071 which show for 10 keV O+ strong increase under the duskward IMF indicated by the clock angle in the inner magnetosphere.

12. lines 251-268, another reason can be that Bouhram et al., 2005 have used some-what different energy range for O+ observations.

13. line 276, magenetopause –> magnetopause

14. line 279, have –> has

15. line 287, dominated occurring –> occurs predominantly

18. lines 296-297, I would change this sentence to "The reconnection rate is likely will be reduced by the mass-loading but not suppressed at the magnetopause [Fuselier et al., 2019].

19. Figure 1, caption, "The three components of the IMF, Bx, By, Bz..."

20. Figure 2, I would indicate on the plot contamination. In the caption, line 481 (k)–>(l)

---

## Referee Comment (RC3) · Anonymous Referee #3 · 27 Aug 2019

This manuscript provides a statistical analysis of the energetic O+ density, O+ flux, and ratio of O+/H+ observed by MMS at the duskside magnetopause during intense substorms. The paper presents new results which are of interest to the scientific community, however there needs to be greater clarification on much of the statistical methods and conclusions. The results have potential to be published in Annales Geophsicae with consideration and adequate response to the following comments and suggestions.

Comments: Lines 90-95: There is a lot of information leading up to this point in the introduction, however with the lines preceding and in this paragraph itself, it is unclear what is not well understood and how/what this paper will provide to answers to. Currently, the introduction reads as a quite thorough list of previous studies, but it is not readily apparent how they string together, and what they are necessarily building up to. I would suggest stating what the paper will study before this point and tailoring the introduction to build off of that somewhat, because at this point as a reader it is still unclear.

Lines 128-150: HPCA & FPI fluxes are in differential flux and energy flux units. Is there a benefit in having their fluxes in different units? If they are to remain, a point should be included in the text that the units are different.

Lines 128-150: The HPCA flux in panels i-l have artificial striping every 4 energy bins due to way HPCA determines the count rate over 4 energy channels in survey mode. It would be best to correct this, however, describing the artificial striping would also be sufficient. I am also not certain that these HPCA fluxes are actually omni-directional as they do not appear to be half-spin averaged, please verify.

Lines 134-137: Please describe where the FPI/HPCA moments shown come from. This is quite important since the majority of the results presented are dependent on these moments.

Figures 1-2: I would suggest using these two figures to establish the criteria for the statistical study. In my opinion, more text should be added that describes a greater context for these 2 figures inclusion. Either establishing points that lend themselves to the paper's conclusion and/or use the figure to establish conditions for the statistical study.

Lines 176-181: This is one of the more major comments on the paper. The current description of the event selection criteria is not sufficient. Interpretation of a statistical study is almost entirely dependent on understanding how the statistical study is conducted. It is currently not clear what the criteria for event selection is. Is it any MP crossing with AE > 500? Why was 500 chosen as a threshold in AE (i.e. stats are somewhat low, would AE > 300 or 400 provide more events and still be "intense"?)

How exactly is the magnetopause boundary layer determined? Is there any consideration for if the substorm is during a storm or the 1st/2nd/3rd in a series of substorms? Specifically, how are substorm phases determined? What is meant by the mean value of the flux (over a range of energies, one energy)? How long were the average events? Please provide greater context for the choices of criteria used in this study.

Lines 179-180: One of the main points from this paper is that the high-density O+ can be transported from the nightside tail to the magnetopause where it is observed. Please discuss any effect (or lack thereof) of using OMNI solar wind values at the bow shock to correlate with observations of high O+ density which is being driven by processes which invariably take some amount of time to occur.

Lines 203-205: With the decimation of HPCA fluxes during survey mode, the count rate is recorded/distributed over 3-4 energy channels. With this in mind, is it appropriate to describe the comparisons of the flux as being over such a small energy range, since the flux/count rate could have been dominated by a nearby energy channel? Potentially, it would be more accurate to re-bin the HPCA flux into 16 energy channels instead of 63, and compare the >1 keV flux levels of these larger energy bins. Please discuss, currently it seems a bit misleading to describe the flux as being over such a narrow energy range.

Lines 231-236: Here it is stated that, "the maximum number density of energetic O+ at the dusk flank magnetopause is during the intense substorms recovery phase under the southward IMF. But the maximum ratio of n(O+)/n(H+) at the dusk flank magnetopause is during intense substorm recovery phase under the northward IMF. IMF Bz seems play a minor role in O+ 235 abundance at the dusk flank magnetopause during intense substorm." It is not clear from the data as it is presented that this is true. The density ratio is of course dependent on O+ and H+ (which can come from the ionosphere and the sw). Comparing Figures 4a and 5a, it is not clear to me by eye that n(O+) is more dependent on By than Bz. It very well may be, but it is not readily apparent. Thus, is the density ratio difference actually from O+ or H+? Additionally, only 6 of

the events in the study have a Bz > 0. This is notable, as Bz not being random does have an impact on the events. Thus, from this study it appears that Bz does play a role in the events being studied.

Lines 241-242: "number density ratio at the dusk flank magnetopause during intense substorms have a weak correlation with the solar wind dynamic pressure." Can you quantify this correlation? In general, there are a lot of points currently that are driven from visual inspection of very scattered plots, when greater statistical rigor perhaps could be applied.

Figures 4-7: The captions of the figures mention that the 95% confidence intervals are shown. Please mention this in the text and describe how it is calculated.

Very minor comments: Lines 103-106: Please explicitly state that FPI does not discriminate between different ion species.

Line 107: Strictly speaking, HPCA measures up to $\sim$40 keV/q (thus for He++ this gets up towards 80 keV).

Line 116: The authors might as well finish this thought, that this is due to spacecraft separation/scales of particle motion.

Line 296: Fuselise et al. should be Fuselier.

Lines 304-306: I would re-phrase this sentence. It is a minor distinction, but it currently reads as if you have studied energetic O+ across the entire magnetopause during substroms and found that the most prevalent region of O+ is the dusk flank during the recovery phase. Whereas, it should be more like, "Observations of energetic O+ at the dusk flank magnetopause during substorms are mainly found within the recovery phase."

---

## Author Comment (AC2) · 21 Sep 2019

Dear reviewer: We are very grateful to your comments for the manuscript and thanks for carefully evaluating this manuscript. According to your advice, we amended the relevant part ofthe manuscript. The one-to-one responses to your comments are the following.

Major comments Comments 1: One of the conclusions of the manuscript is that particles are transported from the tail towards the dayside. To make such a conclusion more rigid one should show the anisotropy of the particle distributions, which would indicated that particles move from the tail towards the dayside. The oxygen ions could

also come from other sources such as inner magnetosphere (filled directly from the nightside aurora into the ring current), from the diamagnetic cavities/cusp (e.g. Slapak et al., Ann. Geophys. 2013,10.5194/angeo-31-1005-2013).

Response: Thank you for pointing this out. Yes, the oxygen ions at the dayside LLBL have many sources such as the ring current in the inner magnetosphere, the high latitude auroral region and the cusp. Our paper focus on the oxygen ions in the dayside LLBL during intense substorms with AE >500nT. Previous research work has reported that the oxygen ions transferred faster into the ring current in the inner magnetosphe and then they are decayed at the dayside magnetopause under southward IMF or with their large gyroradius effect [e.g., Zong et al., 2001]. Under intense geomagnetic activities such as intense substorms and storms, the oxygen ions from the nightside aurora along the plasma sheet or plasma sheet boundary layer can be fast transferred into the near-Earth magnetotial and then injected into the ring current [e.g., Duan et al.,2017 JGR; Yu and Ridley,2013 JGR]. Recently, Kronberg et al. [2014] reported that the oxygen ions distribution was really anistropic at the dawn-dusk equator plane. Our observation result is consistent with their report. I have to admit making such a conclusion is not rigid. Because we can't exclude other origins. I corrected this expression in my revised paper.

Kronberg, E. A., Ashour-Abdalla, M., Dandouras, I., Delcourt, D. C., Grigorenko, E. E., Kistler, L. M.,...Zelenyi, L. M. (2014). Circulation of heavy ions and their dynamical effects in the magnetosphere: Recent observations and models. Space Science Reviews, 184(1-4), 173–235. https://doi.org/10.1007/s11214-014-0104-0 Yu, Y., and A. J. Ridley (2013), Exploring the influence of ionospheric O+ outflow on magnetospheric dynamics: dependence on the source location, J. Geophys. Res. Space Physics, 118, 1711–1722, doi:10.1029/2012JA018411 Zong, Q.-G., B. Wilken, S. Y. Fu, T. A. Fritz, A. Korth, N. Hasebe, D. J. Williams, and Z.-Y. Pu (2001), Ring current oxygen ions escaping into the magnetosheath, J. Geophys. Res., 106(A11), 25,541–25,556.

Comments 2: I am not sure if one could make firm conclusions about dependence on

the IMF Bz,if from 31 events only 6 events were observed during northward IMF. On my opinionthe statistics is too poor for that.

Response: Thank you for the comment. The events of energetic oxygen ions at the dayside LLBL during intense substorms in our studies are chosen from MMS Phase 1a and 1b. Because there are limited number events of intense substorms when MMS passes through the low latitude magnetopause duing the Phase 1a and 1b. The intense substorms are usually occurring duing the southward IMF Bz . Our work presents 31 intense substorms events with 25 events under the sourthward IMF Bz and only 6 events under the northward IMF Bz. This is consistent with the usually external condition of intense substorms [Lyons et al., 2005; Hsu and McPherron, 2003]. On the other hand, we will present a long time periods of MMS observations on the dayside LLBL to present large number events for our following statistics research work. This conclusion will be substitute with a more rigid expression in my revised manuscript. —>"The O+ abundance dependence on IMF Bz is not prominent at the dusk flank magnetopause during intense substorm in our statistical results". Hsu, T.-S., and R. L. McPherron (2003), Occurrence frequencies of IMF triggered and nontriggered substorms, J. Geophys. Res., 108(A7), 1307, doi:10.1029/2002JA009442. Lyons, L. R., D.-Y. Lee, C.-P. Wang, and S. B. Mende (2005), Global auroral responses to abrupt solar wind changes: Dynamic pressure, substorm, and null events, J. Geophys. Res., 110, A08208, doi:10.1029/2005JA011089.

Comments 3: The "intense substorms" are discussed in this study. Were these substorms associated with magnetic storms? Or these are pure substorm events? What is the reason for choosing intense substorms? Inlcuding other substorms may increase the statistics on the IMF dependence.

Response: Thanks for the referee's kind suggestion. In this statistical study, 31 magnetopause crossing events during intense substorm (AE>500 nT) were selected. Among them, there are 4 events during the non-storm time (Dst> -25 nT) and 27 events during the storm time (Dst< -25 nT). There are three resons that we focus on investigating

the characteristics of energetic oxygen ions at the duskside low latitude boundary layer during intense substorms. Firstly, previous studies have reported that the number density and energy flux of oxygen ions in the magnetosphere both increase during magnetic activities, such as intense substorm and storms [e.g.,Daglis et al.,1994;Kronberg et al.,2014]. Second, the characteristics of energetic oxygen ions at the dayside low latitude boundary layer during intense substorms have seldom be reported till now. Oxygen ions play a significant role in the energy and mass transport in the coupling process of the solar wind-magnetosphere-ionosphere during intense substorms. Third, MMS project can provide a good chance to investigate the features of energetic oxygen ions in the dayside low latitude boundary layer. The previous spacecraft observations provided significant results of oxygen ions mainly focusing on the middle and high latitude region, such as Cluster [e.g., Nilsson et al.,2006;Slapak et al.,2011]. Thus, our investigation can provide new results in the dayside LLBL.

Daglis, I. A., Livi, S., Sarris, E. T., & Wilken, B. (1994). Energy density of ionospheric and solar wind origin ions in the near-Earth magnetotail during substorms. Journal of Geophysical Research, 99(A4), 5691–5703. https://doi.org/10.1029/93JA02772 Kronberg, E. A., Ashour-Abdalla, M., Dandouras, I., Delcourt, D. C., Grigorenko, E. E., Kistler, L. M.,...Zelenyi, L. M. (2014). Circulation of heavy ions and their dynamical effects in the magnetosphere: Recent observations and models. Space Science Reviews, 184(1-4), 173–235, doi:10.1007/s11214-014-0104-0. Nilsson, H., et al. (2006), Characteristics of high altitude oxygen ion energization and outflow as observed by Cluster: A statistical study, Ann. Geophys., 24, 1099–1112. Slapak, R., Nilsson, H., Waara, M., André, M., Stenberg, G., and Barghouthi, I. A.( 2011), O+ heating associated with strong wave activity in the high altitude cusp and mantle, Ann. Geophys., 29, 931–944, doi:10.5194/angeo-29-931-2011

Comments 4: Introduction, first two paragraphs can be merged as they contain repeating information about acceleration during dipolarizations. The second paragraph is not completely logical. It would make more sense to describe acceleration of O+ starting

from the polar region, then lobe, dipolarizations and then discuss drift. The sentence in lines 43-45 discussing acceleration of electrons during dipolarizations is not really needed as there is a number of references about acceleration of oxygen during dipolarizations in lines 29-47 and the whole text is about O+.

Response: Thanks for the referee's kind advice. As you suggested, I merged the first two paragraphs to make the introduction more logical and concise.The part of revision can be found in Line 32-50 in revised manuscript.

Comments 5: Lines 90-91, "At present, O+ near the dayside low-latitude magnetopause during substorm expansion phase and recovery phase are still not understood" –> What exactly do you mean under not understood? Which scientific questions are still open? Which questions do you try to answer?

Response: Thank you for these comments. Actually, what we want to know is how the O+ abundance (O+/H+) in the dusk flank magnetopause varies on AE index and solar wind conditions (e.g. IMF By, IMF Bz, and solar wind dynamic pressure) during the intense substorm (AE >500 nT).The relevant description is revised in Line 87-94.

Comments 6: Lines 91-93, there is paper by Luo et al., JGR, 2017, 10.1002/2016JA023471, in which the energization of O+ at the dayside is discussed. The study also discusses asymmetries of the energetic oxygen due to IMF By and Bz directions. Both IMF By and Bz influence the oxygen abundance at higher energies. However, this is large statistical study and not only cases for the intense substorms. This can be discussed.

Response: That would be great. We discussed Luo et al.,(2017) results in my revised manuscript (see Line 337-340). Recently, Using energetic ion composition data at the low latitude dayside magnetopause measured by Magnetospheric Multiscale (MMS) satellites, we study the response of O+abundance (O+/H+) to the both IMF Byand Bz and not only cases for the intense substorms. We found that they indeed influence the oxygen abundance even at lower energies (1-40keV) and more significant duskside asymmetry of O+ under southward IMF with positive IMF By.These results are consistent with those of Luo et al.,(2017).

Comments 7: Lines 125-126, 130-131, 180-181, please provide a more precise definition of the substorm onset and recovery phase. For example in paper by Newell and Gjerloev,JGR, 2011, 10.1029/2011JA016779, is a nice example on how to define substorm onset, also using more precise SML index available at the SuperMAG. I do not think that definition when "AE index significantly increases" is a precise one. I do not think that one should provide twice the information about substorm onset in lines 125-126 and 130-131. I would remove the second sentence.

Response: Thanks for constructive comments and nice recommendation. We have added a more precise definition of the substorm onset, expansion phase and recovery phase in our revised manuscript.The second information about substorm phase description in lines125-126 and 130-131 has been removed.We added AU, AL index in Figure 1 to help us identify the phase of a substorm. First, we determined the time interval of the magnetopause boundary layer crossings in each event. Then, we find out how the substorm indices change during that interval from the OMNI data. As Figure 1 shown, the time interval of the magnetopause boundary layer crossing is indicated by the two blue dashed lines. As we know, the AE index is defined as AE=AU-AL. Generally,the substorm onset time is characteristic by the AL index starts to significantly decrease and the AE index significantly increase. During the substorm expansion phase, the AL index will decrease significantly. The interval of the AL index decrease from onset to its minimum is defined as the substorm expansion phase. Then it starts to increase and the interval of the AL index increase from the minimum to the quiet time level is regarded as the substorm recovery phase. In our event, the MMS4 crossed the magnetopause boundary layer from 15:25:10 to 15:36:50 UT on 3 October 2015. From Figure 1f, the AL index reached its minimum $\sim$-750 nT and AE index reach the peak $\sim$1000 nT at about 15:20 UT, then it started to increase to $\sim$ -200 nT at the rest time of interest. So the magnetopause boundary layer crossing occurred during the intense

substorm recovery phase. (see Line 133-146 in revised manuscript)

Comments 8: Lines 179-180, actual observations of the IMF and solar wind dynamic pressure could be used directly from the MMS observations at the magnetopause crossings.This would be much more precise.

Response: Getting the much more precise IMF and solar wind dynamic pressure would be better. When the IMF passes through the bow shock, its direction would be changed in the magnetosheath. We will compare the IMF and solar wind dynamic pressure directly from the MMS with those from OMNI data in the detailed events analysis. Then we will choose the more precise data.

Comments 9: Lines 277-278, For higher energies the larger statistics one can clearly see that the stronger duskward asymmetry in the plasma sheet and the dayside magnetosphere is observed under the southward IMF, e.g. Luo et al., JGR, 2017. One should mention that no influence of IMF Bz is observed in case of the energies below 40 keV and for 31 intense substorm events.

Response: Thanks for your constructive suggestion.We agree with your comments. Recently, We used energetic ion composition data at the low latitude dayside magnetopause measured by Magnetospheric Multiscale (MMS) satellites, we study the response of O+ abundance to IMF Bz. The O+ abundance showing stronger duskward asymmetry in the magnetopause also be found in our study, which is consistent with Luo et al.2017 result. As you suggested, no influence of IMF Bz is only observed in case of the energies below 40 keV and for 31 intense substorm events. This description is more accurate for our results and relevant revision can be found in Line 25-26, 371-372 in revised manuscript.

Comments 10: Lines 286-287, the energetic O+ occurs predominantly under southward IMF. Here I would say that it was chosen to be like this. Choosing the intense substorms one increases the probability of observing the southward IMF quite significantly. This also contradicts to statement in the lines 277-278, that IMF Bz does not

influence abundance of O+ at the magnetopause. There is not enough provided data to conclude so.By increasing the number of events under the northward IMF one may see a different picture. One can see pretty nice trend in Figure 6b, that the abundance is increasing with the decrease of IMF Bz at least for the expansion phase. Generally on my opinion there is not enough statistics in this study to make conclusions about IMF dependence. One should expand the statistics

Response: Thanks for your valuable comments. We agree with you that choosing the intense substorms one increase the probility of observing the southward IMF significantly. Yes, we can see the pretty nice trend that O+abundance increase with the IMF Bz increase during the intense substorm expansion phase. Due to not enough statistical events, some conclusions may be not convincing. As the MMS operating longer, more magnetopause crossing during intense substorm will be detected. It will be helpful.

Comments 11: Lines 304-306, this conclusion is not supported by the observations. Just looking at the scatter points of the number density, I do not see a statistically significant difference between these two phases. One should either show fits to those points or bin them according to some parameters and show that the difference is significant.

Response: Thanks for your comments. At the beginning of this study, we focus on the response of O+ abundance on the geomagnetic activity and solar wind conditions during intense substorms. Because the magnetosphere has the different dynamics in the near-Earth space during the different phase of intense substorms, especially in substorm expansion phase and recovery phase. We investigate variations of energetic O+ density at the duskside magnetopause boundary layer duing intense substorms by MMS phase 1a and 1b data. Due to the number of events are limited (only 9 events during expansion phase), we don't think it makes sense to fit those points or bin then according to some parameters. As the MMS operating longer, more magnetopause crossing during intense substorm will be detected. It will be helpful. Our selecting events we drawn our summary on the energetic O+ density as description in the last

part of our manuscript. In generally, the O+ in the magnetosphere are origin from the ionosphere and transferred into the different magnetosphere region during magnetic activities. A excellent review paper of this aspect has been reported by Keika et al., [2013]. Our new results from MMS data provide another support of previous studies. Keika, K., L. M. Kistler, and P. C. Brandt (2013), Energization of O+ ions in the Earth's inner magnetosphere and the effects on ring current buildup: A review of previous observations and possible mechanisms, J. Geophys. Res. SpacePhysics, 118, 4441–4464, doi:10.1002/jgra.50371.

Comments 12: lines 313-315, energetic oxygen ions also indicate the transport at the dayside magnetosphere (e.g. Liao et. al, JGR, 2010, 10.1029/2010JA015613). These different transports are hard to distinguish (e.g. Luo et al., JGR, 2017).

Response: Thanks for your comments and paper recommendation. Liao et al.,(2010) JGR and Luo et al.,(2017) JGR are both cited in our revised manuscript. The different transports of oxygen ions from the ionosphere to different part of the magnetosphere are significant and interest. It is outside the focus of our manuscript. We will investigate this issue with conjunction observations by multiple spacecraft in different magnetosphere locations.

Comments 13: Figures 4-7, just looking at the scatter plots it is hard to make certain conclusions.One should either bin the points to show the average trend or fit them with some dependences and increase the number of events.

Response: Thanks for your nice suggestions. I have binned the points to show the average trend before submitted this manuscript. As you said,the number of events is too low, so the trend is not obviously or has low credibility and we abandoned this method. Recently, Using energetic ion composition data at the low latitude dayside magnetopause measured by Magnetospheric Multiscale (MMS) satellites, we study the response of H+, O+density and their ratio to the geomagnetic activity (indicated by SYM-H index) and solar wind conditions (including interplanetary magnetic field (IMF)

By, IMF Bz and solar wind dynamic pressure). In this study, we bin the points due to enough events. Our new manuscript has been submitted to the JGR.

Minor comments: 1. Line 19: What is the energy range of the oxygen observations used in this study?Please indicate the upper energy limit in the abstract. This is important to know whenassessing the number densities.

Response: In this study, only the O+ at energies from 1 to 40 keV measured by HPCA are used.The upper energy limit of HPCAis 40 keV.This information is added to the abstract. (see Line 16)

2. Line 45: I did not find the reference to Lui et al., 1999 in the reference list. 3. Line 47: "during activity geomagnetic disturbance" –> "during disturbed geomagneticactivity" 4. Line 55: "[e.g. Yau and Andre, 1997]. And then..." –> "[e.g. Yau and Andre, 1997].Then..." 5. Line 85: please remove one "However".

Responseto comments from 2-5:The above expression errors have been checked and corrected. The missing reference has been added into the revised manuscript.

6. Lines 106-107:Does HPCA distinguish between O+, N+ and C+? Or what measuredis actually the CNO group?

Response: "The HPCA is a time-of-flight (TOF) mass spectrometer designedto measure the velocitydistributions of the four ion species (H+ , He++ , He+ and O+ ) knownto be important in the reconnection process. The measurement technique is based on a combinationof electrostatic energy-angle analysis with time-of-flight velocity analysis. The resultis an accurate determination of the velocity distributions of the individual ion species.In order to meet the stringent scientific requirements of the MMS mission, the HPCA incorporatesthree new technologies. The first extends counting rate dynamic range by employinga novel radio frequency mass filter that allows minor species such as He++ and O+ to bemeasured accurately in the presence of intense proton fluxes found in the dayside magnetopause.The second ensures that TOF processing rates

are high enough to overlap with thelow end of the RF dynamic range, while the third enhances ion mass resolution.

During each energy scan a data set consisting of 63 TOF spectra × 512 TOF bins × 16elevations isaccumulated and histogrammed. The resulting TOF spectra are then parsed intofive bins that define the ion species H+, He++, He+, O+ and background (Fig. 20). The redportions of the spectrum in Fig. 20 indicate typical species boundaries. Since ion times-offlightare both mass and energy dependent the range of TOF limits for each species changeswith energy (Fig. 31)". (the Figure and description are cited from Young, D. T., Burch, J. L., Gomez, R. G., De Los Santos, A., Miller, G. P., Wilson, P., et al. (2016). Hot Plasma Composition Analyzer for the Magnetospheric Multiscale Mission. Space Science Reviews, 199(1–4), 407–470, doi:10.1007/s11214-014-0119-6.).

As for this interesting question, I specially contacted the HPAC PI (Stephen Fuselier), he replied me "I'm working right now to see if we can see C+ and possibly N+ in the mass spectra. They would not appear as a separate mass peak because of straggling in the foil. I'm not sure if we can even tell if they are there. What we bring to ground and call O+ could contain substantial N+. The C+ peak would probably be at a lower time-of-flight than what we bring to ground, but you could safely say that what we call O+ could be N+O+."

7. Line 122: "At the beginning of the time interval, the solar wind dynamic pressure..."–>The dynamic pressure is only at the begging of the time interval about 2 nPa. 8. Lines 124-125: I would change to " These solar wind conditions led to the intensesubstorm (AE>500 nT). 9. Lines 148-150: Figure 2, I would say that the fluxes at energies below 2 keV inFigure 2j is also contamination. This should be mentioned also in Figure caption andeven better when it is indicated on plot itself.

Responseto comments from 7-9: Thanks for referee's valuable suggestion. The minor comments 7 and 8 have been corrected in my manuscript. The red box indicating

the O+ contamination from high proton fluxes was plotted in figure 2j and relevant description was mentioned in the Figure 2 caption (see Line 160 -161).

10. Line 195:I would remove "On the other hand".

Response: This is a common colloquial expression. We removed it.

11. Lines 221-224:These results also agree with Kronberg et al., JGR, 2012,10.1029/2012JA018071 which showed for 10 keV O+ strong increasing under the duskwardIMF indicated by the clock angle in the inner magnetosphere.

Response: That would be great. We cited this paper in the relevant part of revised manuscript to increase valid of our results.

12. Lines 251-268: another reason can be that Bouhram et al., 2005 have some-what different energy range for O+ observations.

Response: Yes, We agree with you. I can't exclude the reason that Bouhram et al., 2005 used somewhat different energy range for O+ observations.In this study, the O+ density calculated using HPCA distribution functions at energies from 1 to 40 keV, but Bouhram et al., (2005) used CODIF distribution functionsat energies from 3 to 40 keV to contamination from high H+ fluxes. This contrast study is not rigid in this study. We removed the relevant part in my revisited manuscript.

13. Line 276:magenetopause –> magnetopause 14. Line 279: have –> has 15.Line 287: dominated occurring –> occurs predominantly 17. Lines 296-297: I would change this sentence to "The reconnection rate is likely willbe reduced by the mass-loading but not suppressed at the magnetopause [Fuselier etal., 2019]. 17. Figure 1, caption, "The three components of the IMF, Bx, By, Bz..." 18. Figure 2, I would indicate on the plot contamination. In the caption, line 481 (k)–>(l).

Response to comments from 13-18: Thanks for referee'scarefully evaluating this paper and important suggestions. We have revised above errors and plotted the red box indicating the O+ contamination from high proton fluxes in figure 2j. The other spelling

and syntax errors have been checked and corrected. We acknowledge the reviewer's comments and suggestions very much, which are valuable in improving the quality of our manuscript.

_Interactive comment on Ann. Geophys. Discuss., https://doi.org/10.5194/angeo-2019-90, 2019._

OMNI

**Fig. 1.** Fig.1: The three components IMF Bx, By, Bz, solar wind dynamic pressure, as well as AU, AL, and AE index from CDAweb OMNI data.

[Figure]

**Fig. 2.** Fig.2 The energetic O+ is observed at the magnetopause during an intense substorm on 03 October 2015 by MMS 4.

**Fig. 20** TOF spectrum for four
ion species and background ($H_2^+$
is a substitute for $He^{++}$ and $N^+$
is a substitute for $O^+$). *Red areas
demarcate bins that define ion
species and background. The
peak at ~200 ns corresponds
to $N_2^+$*

[Figure]

**Fig. 3.** Figure 20

**Fig. 31** TOF boundaries as a function of energy for four ion species plus background

[Figure]

**Fig. 4.** Figure 30

---

## Author Comment (AC3) · 21 Sep 2019

Dear reviewer: We are very grateful to your comments for the manuscript and thanks for carefully evaluating our manuscript. According to your advice, we amended the relevant part of the manuscript. Responses to your comments are below point by point.

Comments 1: Lines 90-95: There is a lot of information leading up to this point in the introduction, however with the lines preceding and in this paragraph itself, it is unclear what is not well understood and how/what this paper will provide to answers to. Currently, the introduction reads as a quite thorough list of previous studies, but it

[Figure]

is not readily apparent how they string together, and what they are necessarily building up to. I would suggest stating what the paper will study before this point and tailoring the introduction to build off of that somewhat, because at this point as a reader it is still unclear.

Response: Thanks for the referee's kind advice. As you suggested, we did some revisions in our revised manuscript. We merged the first two paragraphs to make the introduction more logical and concise. First, we describe acceleration of O+ starting from the polar region, then lobe, near-Earth plasma sheet and then discuss drift, stressing the importance of O+ during the intense substorms. Second, we describe the O+ behaviour in the magnetopause. Third, we referred the O+ density dawn-dusk asymmetry in the magnetopause. Finally, we describe the questions what this paper try to answer. As the follow describing: "At present, variations of O+ abundance (O+/H+)in the dusk flank magnetopause during intense substorms (AE >500 nT) on AE index and solar wind conditions (e.g. IMF By, IMF Bz and solar wind dynamic pressure) are still not understood. Previous studies of O+ during intense substorms mainly focused on O+ energizations in the NEPS in the magnetotail (e.g.,Duan et al., 2017; Nosé et al., 2000; Ohtani et al., 2011).At present, The Magnetospheric Multiscale (MMS) mission gives us an opportunity to focus on the O+ in the low latitude dayside magnetopause region. In this study, we mainly investigate statistical features of energetic O+ in the dusk flank magnetopause varying on AE index and solar wind conditions (e.g. IMF By, IMF Bz and solar wind dynamic pressure) during the intense substorms.

Comments 2: Lines 128-150: HPCA & FPI fluxes are in differential flux and energy flux units. Is there a benefit in having their fluxes in different units? If they are to remain, a point should be included in the text that the units are different.

Response: Thanks for the referee's kind suggestion. We described the HPCA and FPI fluxes having different units in our revised manuscript. Figure 2g and 2h show the electron omnidirectional differential energy fluxes and ion omnidirectional differential energy fluxes, respectively. Figure 2i to 2l present the differential particle fluxes of H+,

O+, He+, He++, respectively.

Comments 3: Lines 128-150: The HPCA flux in panels i-l have artificial striping every 4 energy bins due to way HPCA determines the count rate over 4 energy channels in survey mode. It would be best to correct this, however, describing the artificial striping would also be sufficient. I am also not certain that these HPCA fluxes are actually omni-directional as they do not appear to be half-spin averaged, please verify.

Response: Thanks for your important comments. Figure 2i to 2l present the differential particle fluxes of H+, O+, He+, He++, respectively. They are actually not omni-directional and not half-spin averaged. We corrected this description in our revised manuscript. These differential particle fluxes of H+, O+, He+, He++ calculations are used The Space Physics Environment Data Analysis System (SPEDAS) software package. More details about SPEDAS can be found in Angelopoulos et al. (2019) and cited as (Angelopoulos, V., Cruce, P., Drozdov, A. et al. Space Sci Rev (2019) 215: 9. https://doi.org/10.1007/s11214-018-0576-4). We also cited this paper in our revised manuscript (see Line 111-113).

Comments 4: Lines 134-137: Please describe where the FPI/HPCA moments shown come from. This is quite important since the majority of the results presented are dependent on these moments.

Response: Thanks for the referee's kind advice. We have clarified where the FPI/HPCA moments shown come from. We have added detailed information about moments as the description in Figure 2. The plasma moments (e.g. Ion parallel and perpendicular temperatures, ion, and electron number densities and ion velocity) from FPI shown in Figure 2c-2e are all from MMS L2 data products. They are default moments calculated over the full FPI energy range from 10 eV to 30 keV. But the O+ density shown in Figure 2f is recalculated from HPCA distribution functions in the range of energies from 1 to 40 keV. From the O+ fluxes shown in Figure 2j, there still exist a large number of fluxes below 1 keV in the magnetosheath. This part of O+ fluxes is fake and contamination

from high proton fluxes. So we consider the number density of O+ with energies from 1 to 40 keV. It is more appropriate to represent the true O+ in the magnetopause. While the H+ density (over the full HPCA energy range) from L2 data products are used in Figure 2f.

Comments 5: Figures 1-2: I would suggest using these two figures to establish the criteria for the statistical study. In my opinion, more text should be added that describes a greater context for these 2 figures inclusion. Either establishing points that lend themselves to the paper's conclusion and/or use the figure to establish conditions for the statistical study.

Response: Thanks for your nice comments. In this statistical study, First, we identified the magnetopause crossing event (complete magnetopause crossing from the magnetosheath to the magnetosphere, vice versa) during phase 1 from the summary plot in https://lasp.colorado.edu/mms/sdc/public/plots/.Then we plotted the more detailed overview of these eventsto identify the magnetopause boundary layers, as Figure 2 shown. Figure 2 was mainly used to determine the magnetopause boundary layer crossing interval. Only events that AE index larger than 500 nT during the magnetopause boundary layer crossings interval were selected. Finally, the mean value of the H+, O+ density and their fluxes shown in Figure 2 were calculated in that interval. Correspondingly, the AE index, IMF By, Bz and solar wind dynamic pressure from the OMNI data system shown in Figure 1 were also averaged during that interval. Figure 1 mainly provided the corresponding solar wind conditions and AE index.

Comments 6: Lines 176-181: This is one of the more major comments on the paper. The current description of the event selection criteria is not sufficient. Interpretation of a statistical study is almost entirely dependent on understanding how the statistical study is conducted. It is currently not clear what the criteria for event selection is. Is it any MP crossing with AE > 500? Why was 500 chosen as a threshold in AE (i.e. stats are somewhat low, would AE > 300 or 400 provide more events and still be "intense"?).

[Figure]

Response: Thanks for your valuable comment. The magnetopause crossing event in our statistical study all during the intense substorm (AE > 500nT). How we chosen these events? First, we identified the magnetopause crossing event (complete magnetopause crossing from the magnetosheath to the magnetosphere, vice versa) during phase 1 from the summary plot in https://lasp.colorado.edu/mms/sdc/public/plots/. Then we plotted the more detailed overview of these events, as figure 2 shown. Next, we identiïfied the magnetopause boundary layers primarily through electron and ions energy fluxes and moments. Only events that AE index larger than 500 nT during the magnetopause boundary layer crossings are selected. The reason why we choose intense substorm with AE >500 nT is based on the result from Daglis et al (1994) (Figure 6 in this reference, as shown in below Figure 3). A number of previous studies have demonstrated that the O+ abundance relates to the substorm process. (Lennartsson and Shelley, 1986) pointed out that the ion composition has a large variance at substorm. During the intense disturbed conditions (AE~1000nT), the increase in the O+ energy density is strongest around local midnight where O+ become the most numerous ion. O+ energy density has a great correlation with the AE index in the near-Earth plasma sheet (NEPS)was also founded by(Daglis et al., 1994). During the intense substorm expansion phase, O+ energy density explosively increases with AE index in the range of larger than 500nT. The previous researches of oxygen ions during intense substorms are mainly focus on the nightside near-Earth plasma sheet (NEPS). Thus, we want to know whether the O+ abundance in the dusk flank magnetopause varies on AE index and solar wind conditions during the intense substorm and how it changes to the above parameters. Characteristics of Oxygen ions in the high latitude polar region and near-Earth magnetosphere during intense magnetic activities have been investigated deeply and widely. But O+ abundance in the low latitude dayside magnetopause has seldom report during intense substorms.

Comments 7: How exactly is the magnetopause boundary layer determined? Is there any consideration for if the substorm is during a storm or the 1st/2nd/3rd in a series of substorms? Specifically, how are substorm phases determined? What is meant by

the mean value of the flux (over a range of energies, one energy)? How long were the average events? Please provide greater context for the choices of criteria used in this study.

Response: The magnetopause boundary layers are identified here primarily through plasma fluxes and moments. The low-latitude boundary layer (LLBL) on the magnetospheric side of the magnetopause current layer and the magnetosheath boundary layer (MSBL) on the magnetosheath side of the magnetopause current layer. They can have densities and temperatures between that of the magnetosphere and magnetosheath. At the same time, MP boundary shows the gradient of the energy flux of particles and number density and magnetic field obvious. Ion jets are also signatures of passing through the magnetopause boundary layers. In this study, the separatrix between the magnetosheath and the magnetopause boundary layer is determined by the appearance of the magnetospheric electron, as the first black solid line in Figure 2 shown. Similarily, the separatrix of the magnetosphere and the magnetopause boundary layer is determined by the magnetosheath electron disappearance, as the second solid line in Figure 2 shown. The mean value of the H+, O+ density and their fluxes are calculated in the magnetopause boundary layer. Correspondingly, the AE index, IMF By, Bz and solar wind dynamic pressure from the OMNI data system were averaged during the time interval of magnetopause boundary layer crossing. As Figure 2 shown, the time interval of the magnetopause boundary layer crossing is marked by the two blue dashed lines. As we know, the AE index is defined as AE=AU-AL. Generally, the substorm onset time is characteristic by the AL index starts to significantly decrease and the AE index significantly increase. During the substorm expansion phase, the AL index will decrease significantly. The interval of the AL index decrease from onset to its minimum is defined as the substorm expansion phase. Then it starts to increase and the interval of the AL index increase from the minimum to the quiet time level is regarded as the substorm recovery phase. In our event, the MMS4 crossed the magnetopause boundary layer from 15:25:10 to 15:36:50 UT on 3 October 2015. From Figure 1f, the AL index reached its minimum ∼-750 nT and AE index reach the peak

~1000 nT at about 15:20 UT, then it started to increase to ~ -200 nT at the rest time of interest. The two blue dashed lines indicate the time interval of the magnetopause boundary layer crossing. According to the variation and peak value of the AU, AL and AE index in Figure 1e to 1g.The magnetopause boundary layer crossing occurred during the recovery phase of this intense substorm. The mean value of the flux is over a range of energies close to the typical energy such as 1 keV, 10keV and so on. We didn't consider for if the substorm in during a storm or the 1st/2nd/3rd in a series of substorms. In this statistical study, 31 magnetopause crossing events during intense substorm (AE>500 nT) were selected. Among them, there are 4 events during the non-storm time (Dst> -25 nT) and 27 events during the storm time (Dst< -25 nT).These detailed contexts for choices of criteria used in this study are described in my revised manuscript.

Comments 8: Lines 179-180: One of the main points from this paper is that the high-density O+ can be transported from the nightside tail to the magnetopause where it is observed. Please discuss any effect (or lack thereof) of using OMNI solar wind values at the bow shock to correlate with observations of high O+ density which is being driven by processes which invariably take some amount of time to occur.

Response: Thanks for the referee's good evaluation and kind suggestion. Making such a conclusion is not rigid. I didn't give direct evidence to prove that these O+ are transported from the tail towards the dayside. So, I corrected this expression in my revised paper.

Comments 9: Lines 203-205: With the decimation of HPCA fluxes during survey mode, the count rate is recorded/distributed over 3-4 energy channels. With this in mind, is it appropriate to describe the comparisons of the flux as being over such a small energy range, since the flux/count rate could have been dominated by a nearby energy channel? Potentially, it would be more accurate to re-bin the HPCA flux into 16 energy channels instead of 63, and compare the >1 keV flux levels of these larger energy bins. Please discuss, currently it seems a bit misleading to describe the flux as being over

such a narrow energy range.

Response: Thanks for the referee's nice comment and kind suggestion. The main purpose of calculating the O+/H+ particle fluxes ratio is to study the O+ abundance at different energies on AE index and solar wind conditions (e.g. IMF By, IMF Bz and solar wind dynamic pressure) during the intense substorms. Since the energy range of O+ and H+ in the HPCA are the same. So we directly divide O+ particle fluxes by H+ particle fluxes and mainly concentrate on the ratios.

Comments 10: Lines 231-236: Here it is stated that, "the maximum number density of energetic O+ at the dusk flank magnetopause is during the intense substorms recovery phase under the southward IMF. But the maximum ratio of n(O+)/n(H+) at the dusk flank magnetopause is during intense substorm recovery phase under the northward IMF. IMF Bz seems play a minor role in O+ abundance at the dusk flank magnetopause during intense substorm." It is not clear from the data as it is presented that this is true. The density ratio is of course dependent on O+ and H+ (which can come from the ionosphere and the solar wind). Comparing Figures 4a and 5a, it is not clear to me by eye that n(O+) is more dependent on By than Bz. It very well may be, but it is not readily apparent. Thus, is the density ratio difference actually from O+ or H+? Additionally, only 6 of the events in the study have a Bz > 0. This is notable, as Bz not being random does have an impact on the events. Thus, from this study it appears that Bz does play a role in the events being studied.

Response: Thanks for your valuable comments. The conclusion of "IMF Bz seems play a minor role in O+ abundance at the dusk flank magnetopause during intense substorm." in manuscript is not rigid. It noted that choosing the intense substorms one increase the probability of observing the southward IMF significantly. We found a nice trend that O+ abundance increase with the IMF Bz increase. From Figure 6b, the O+/H+ density ratio show an obvious decrease with IMF Bz increasing from -6 to -3 nT, especially for the expansion phase, as the blue crosses shown. Due to not enough statistical events (only 6 of the events in the study with northward IMF), some conclusions

may be not convincing. As the MMS operating longer, more magnetopause crossing during intense substorm will be detected. It will be helpful. The relevant part has been revised.

Comments 11: Lines 241-242: "number density ratio at the dusk flank magnetopause during intense substorms have a weak correlation with the solar wind dynamic pressure." Can you quantify this correlation? In general, there are a lot of points currently that are driven from visual inspection of very scattered plots, when greater statistical rigor perhaps could be applied.

Response: Thanks for your suggestions. Due to the number of events are limited (only 9 events during expansion phase) and distribution plot is very scattered, we don't think it makes sense to fit those points or bin then according to some parameters. We delete the sentence "number density ratio at the dusk flank magnetopause during intense substorms have a weak correlation with the solar wind dynamic pressure." And substituted by more detailed description "From Figure 7a, the H+ density shows slightly change with the solar wind dynamic pressure. While the O+ density shows a slight decrease with the solar wind dynamic pressure from 1 to 2.5 nPa, more prominent during the intense substorm expansion phase. Then it enhances significantly with the solar wind dynamic pressure from 2.5 to 4.5 nPa. Similarly, the O+/H+ density ratio also decreases slightly with the solar wind dynamic pressure from 1 to 2.5 nPa and then increase obviously from 2.5 to 4.5 nPa."

Comments 12: Figures 4-7: The captions of the figures mention that the 95% confidence intervals are shown. Please mention this in the text and describe how it is calculated.

Response: Thanks for your kind suggestion. How to calculate the Confidence Interval is describe as follow: Step 1: find the number of observations N in the magnetopause boundary layer. Then calculate their mean X and standard deviation S. Step 2: Find the "Z" value for 95% Confidence Interval. For 95% the Z value is 1.960. Step 3: use

that Z in this formula for the Confidence Interval (X ) $\pm Z^*$ S/$\sqrt{}$N Where: X is the mean; Z is the chosen Z-value from the table above; S is the standard deviation; N is the number of observations;

Very minorcomments:

Lines 103-106: Please explicitly state that FPI does not discriminate between different ion species.

Response: thanks for your kind suggestion. We added the "FPI does not discriminate between different ion species" in the Line 103-104.

Line 107: Strictly speaking, HPCA measures up to40 keV/q (thus for He++ this gets up towards 80 keV).

Response: Thanks for you carefully evaluate this manuscript. We agree with you, the HPCA maximum measurement for energy per charge is 40keV/q.

Line 116: The authors might as well finish this thought, that this is due to spacecraft separation/scales of particle motion.

Response: Thanks for your nice suggestion. We added this sentence "this is due to spacecraft separation/scales of particle motion." into the Line 117 for finishing this thought.

Line 296: Fuselise et al. should be Fuselier.

Response: Thanks for you carefully evaluating this manuscript and giving important suggestions. We have revised this error. The other spelling and syntax errors have also been checked and corrected.

Lines 304-306: I would re-phrase this sentence. It is a minor distinction, but it currently reads as if you have studied energetic O+ across the entire magnetopause during substroms and found that the most prevalent region of O+ is the dusk flank during the recovery phase. Whereas, it should be more like, "Observations of energetic O+ at

the dusk flank magnetopause during substorms are mainly found within the recovery phase."

Response: Thanks for referee's nice suggestion. We adopted your sentence "Observations of energetic O+ at the dusk flank magnetopause during substorms are mainly found within the recovery phase." to replace before one. We acknowledge the reviewer's comments and suggestions very much, which are valuable in improving the quality of our manuscript.
* * *
[Figure]

**Fig. 1.** The three components IMF Bx, By, Bz, solar wind dynamic pressure, as well as AU, AL, and AE index from CDAweb OMNI data.

[Figure]

**Fig. 2.** The energetic O+ is observed at the magnetopause during an intense substorm on 03 October 2015 by MMS 4.

[Figure]

Figure 6. Same format as Figure 2; all substorms with $AE_{exp} \geq 500$ nT (encircled symbols) are excluded from the fit.

**Fig. 3.** Daglis et al. (1994): Figure 6

---

## Author Response (AR1)

**Reply to reviewer #1**

Dear reviewer:

We are very grateful for your comments about our manuscript. We acknowledge the reviewer's comments and suggestions very much, which are valuable in improving the quality of our manuscript. According to your advice, we amended the relevant part of the manuscript. The one-to-one responses are the following.

**Comment 1:** Line 138: The authors should clarify the information on which instrument dataset was used for each data product. Were the moments shown in Figure 2c-2e recalculated from the FPI distribution functions? Or are they the default moments calculated over the full FPI energy range?

**Response:** Thanks for the referee's kind advice. We should have clarified the dataset information in Figure 2 on which instrument was used. So I added detailed information about dataset what we used to the description of Figure 2. The plasma moments (e.g. Ion parallel and perpendicular temperatures, ion, and electron densities and ion velocity) from FPI shown in Figure 2c-2e are all from MMS L2 data products. They are default moments calculated over the full FPI energy range from 10 eV to 30 keV. But the $O^+$ density shown in Figure 2f is recalculated from HPCA distribution functions at energies from 1 to 40 keV. From the $O^+$ fluxes shown in Figure 2j, there still exists a large number of fluxes below 1 keV in the magnetosheath. This part of $O^+$ fluxes is fake and contamination from high proton fluxes. So we consider the number density of $O^+$ at energies from 1 to 40 keV is more appropriate to represent the true $O^+$ in the magnetopause. While the $H^+$ density (over the full HPCA energy range) from L2 data products are used in Figure 2f. The magnetic and electric fields in GSM are from FGM and EDP, respectively. The last four panels of Figure 2 show the omnidirectional differential fluxes of four individual ion species, $H^+$, $O^+$, $He^+$, and $He^{++}$ measured by HPCA, respectively. See Line 198-207 in "Tracked change" manuscript.

**Comment 2:** Line 140 (Figure 2f): The calculations performed to derive the >1 keV $O^+$ density need to be described to inform the reader how the HPCA energy ranges were specified for those calculations. If a software package was used, then details of the software package and a citation to it should be included. The >1keV $H^+$ density could also be plotted in this figure panel.

**Response:** Thanks for the referee's good suggestion. As mentioned before, the >1 keV $O^+$ density (shown in Figure 2f) recalculated from the HPCA distribution functions at energies from 1 to 40 keV. As your suggestion, I also plotted the $H^+$ density over the full FPI energy range from 10 eV to 40 keV in Figure 2f for better comparison. Because of $H^+$ measurements from HPCA is accurate and the $H^+$ mean energy in the magnetosheath is typically 0.3 keV, so we used the $H^+$ density (over the full HPCA energy range) from L2 data products. These $O^+$ density calculations are used The Space Physics Environment Data Analysis System (SPEDAS) software package. More details about SPEDAS can be found in Angelopoulos et al. (2019) and cited as (Angelopoulos, V., Cruce, P., Drozdov, A. et al. Space Sci Rev (2019) 215: 9. https://doi.org/10.1007/s11214-018-0576-4). We also cited this paper in our revised manuscript, see Line 144-148 and Figure 2f in "Tracked change" manuscript.

[Figure]

**Figure 2.** The energetic $O^+$ are observed at the magnetopause during an intense substorm on 03 October 2015 by MMS 4. From top to bottom are (a) the magnetic field three components, Bx (blue line), By (gree line), Bz (red line) and the total magnitude Bt (black line), (b) the electric field three components, Ex (blue), Ey (gree) and Ez (red), (c) Ion parallel (red) and perpendicular(black) temperatures, (d) The number density of ion (green) and electron (blue), (e) three components of the ion velocity, (f) number density of $H^+$ (over the full HPCA energy range) and $O^+$ (at energies from 1 to 40 keV), (g) electron omnidirectional differential energy fluxes, (h) ion omnidirectional differential energy fluxes, (i) to (l) present omnidirectional differential particle fluxes of $H^+$, $O^+$, $He^+$, and $He^{++}$, respectively. The Geocentric Solar Magnetospheric (GSM) coordinate system is adopted. The thick bars at the top of the panel represent different regions encountered on this magnetopause crossing event. The orange and blue bars represent the magnetosheath and the magnetosphere, respectively. The green bar represents the magnetopause boundary layer. The black horizontal line in figure 2j is at 1 keV and the $O^+$ contamination from high $H^+$ fluxes is indicated by the red box. The FPI data in Figure 2c-e and 2g-h are from FPI L2 data product and in the fast mode.

**Comment 3:** Line 158-164: The magnetopause identification criteria are not very convincing. Recommend carefully defining these criteria, as all statistics are derived based on the magnetopause identification. Recommend the authors review identification criteria used in previous works. For example, Haaland et al. (2016) and (2019) describe magnetopause observations by Cluster and THEMIS: Haaland, S., Reistad, J., Tenfjord, P., Gjerloev, J., Maes, L., DeKeyser, J., Maggiolo,R., Anekallu, C., and Dorville, N. (2014), Characteristics of the flank magnetopause:Cluster observations, J. Geophys. Res. Space Physics, 119, 9019–9037,doi:10.1002/2014JA020539.

Haaland, S., Runov, A., Artemyev, A., & Angelopoulos, V. (2019). Characteristics of the flank magnetopause: THEMIS observations. Journal of Geophysical Research: Space Physics, 124, 3421–3435. https://doi.org/10.1029/2019JA026459. Paschmann et al. (2018) describe magnetopause identification and observations by MMS:

Paschmann, G., Haaland, S. E., Phan, T. D., Sonnerup, B. U. Ö., Burch, J. L., Torbert, R. B., et al. (2018). Large-scale survey of the structure of the dayside magnetopause by MMS. Journal of Geophysical Research: Space Physics, 123, 2018–2033. https://doi.org/10.1002/2017JA025121.

**Response:** Thanks for the referee's kind suggestion and well recommend. We read the papers your recommended and found they did detailed work for magnetopause identification. It deepens my understanding of the flank magnetopause characteristics and helps me identifying the magnetopause more convincing. In this study, we mainly focus on the $O^+$ in the dusk flank magnetopause boundary layer. According to the magnetic field, B is about 40 nT and $O^+$ temperature, T is about 10 keV in this study, we can draw the $O^+$ gyroradius is about 1020 km. From Haaland et al. (2014), the flank magnetopause thickness varies from 150 to 5000 km with a median thickness of around 1150 at dusk. Thus the gyroradius of ten keV $O^+$ is comparable to magnetopause thickness. In that situation, $O^+$ will show the finite Larmor radius effects and the MMS detect partial gyro motion in the magnetopause. For acquiring complete $O^+$ distribution functions, we need to measure $O^+$ in more large spatial scales. So in this study, we focus on the magnetopause boundary layer judgment. The magnetopause boundary layers are identified here primarily through plasma fluxes and moments. The low-latitude boundary layer (LLBL) on the magnetospheric side of the magnetopause current layer and the magnetosheath boundary layer (MSBL) on the magnetosheath side of the magnetopause current layer can have densities and temperatures between that of the magnetosphere and magnetosheath. Ion jets are also signatures of passing through the magnetopause boundary layers. In this study, the separatrix between the magnetosheath and the magnetopause boundary layer is determined by the appearance of the magnetospheric electron. Similarily, the separatrix of the magnetosphere and the magnetopause boundary layer is determined by the magnetosheath electron disappearance. The revised details can be found in Line 221-233 in "Tracked change" manuscript.

**Comment 4:** Line 180: More details are needed to describe how the mean values of the $H^+$ and $O^+$ fluxes and densities were calculated.

**Response:** Thanks for the referee's kind advice. First, we determine the time interval of the magnetopause boundary layer crossings in each event. For example, on 03 October 2015 event, MMS 4 traversed the duskside magnetopause boundary layer from 15:25:10 to 15:36:50 UT judged by the typical characteristics in this region as mentioned before. Then, the $H^+$ and $O^+$ fluxes and densities were average during this time interval. We also give the error bars indicating 90% confidence intervals. We think these mean values represent the $H^+$ and $O^+$ fluxes and densities in the magnetopause boundary layer. See Line 232-233 in "Tracked change" manuscript.

**Comment 5:** Line 184: A more detailed description of how the substorm phase (i.e. expansion phase or recovery phase) was defined based on AE index is needed. The authors should use Figure 1 AE index to aid in their description.

**Response:** Thanks for the referee's suggestion. We should give more details to clarify how we define the substorm phase according to substorm indices, such as AU, AL, and AE index. First, we determined the time interval of the magnetopause boundary layer crossings in each event. Then, see how the substorm indices vary during that interval from the OMNI data. As Figure 1 shown, the time interval of the magnetopause boundary layer crossing is marked by the two blue dashed lines. As we know, the AE index is defined as AE=AU-AL. Generally, the substorm onset time is characteristic by the AL index starts to significantly decrease and the AE index significantly increase. During the substorm expansion phase, the AL index will decrease significantly. The interval of the AL index decrease from onset to its minimum is defined as the substorm expansion phase. Then it starts to increase and the interval of the AL index increase from the minimum to the quiet time level is regarded as the substorm recovery phase. In our event, the MMS4 crossed the magnetopause boundary layer from 15:25:10 to 15:36:50 UT on 3 October 2015. From Figure 1f, the AL index reached its minimum ~-750 nT and AE index reach the peak ~1000 nT at about 15:20 UT, then it started to increase to ~ -200 nT at the rest time of interest. The two blue dashed lines indicate the time interval of the magnetopause boundary layer crossing. According to the variation and peak value of the AU, AL and AE index in Figure 1e to 1g. The magnetopause boundary layer crossing occurred during the recovery phase of this intense substorm. The revised details can be found in Line 166-180 in "Tracked change" manuscript.

[Figure]

**Figure 1.** The three components IMF Bx, By, Bz, solar wind dynamic pressure, as well as AU, AL, and AE index from CDAweb OMNI data.

**Comments 6:** Line 202-209: Several narrow energy ranges used for comparing the $O^+/H^+$ density ratio are noted. It is important to describe for the reader how these energy ranges were used in the density ratio calculations. In addition, a description of why these energy ranges were chosen should be included. Did the authors consider calculating the density ratio for all energies >1 keV instead of calculating the ratios over individual energy ranges? A comparison of density ratios using both methods may be helpful to decide which method to use. Such procedural information on which analysis methodology was chosen could be included in an appendix.

**Response:** Thanks for carefully evaluating this manuscript and kind suggestions. The description in Line 202-209 is not accurate and it appears that the referee has some misunderstanding on what we did. In this study, we calculate the $O^+/H^+$ density ratio (as Figure 4b shown). The $O^+$ density calculated at energies from 1 keV to 40 keV, but the $H^+$ density (over the full HPCA energy range) from L2 data products are used. In order to realize in which individual energy ranges the $O^+$ abundance ($O^+/H^+$) varies obviously on AE index and solar wind parameters. We calculated the particle fluxes ratio at several individual energy ranges (as Figure 4c shown). Since the energy channel range of HPCA for $H^+$

and $O^+$ is the same, so the fluxes ratio are defined as the ratio between their fluxes, We also give the error bars indicating 90% confidence intervals. See Line 280-292 in "Tracked change" manuscript.

**Comments 7:** Line 218-248: Figures 5, 6, 7 all show comparisons of the $O^+/H^+$ density ratio. After addressing the previous comment on Line 202-209 on why separate narrow energy ranges were chosen instead of using a broad energy range, the authors may need to revise panel (b) of these three figures. For example on Line 240: Are the $O^+$ and $H^+$ densities referred to in this section calculated from one of the energy ranges discussed in Line 202-209? Greater detail and explanation are needed.

**Response:** It may be our inaccurate descriptions result in the referee's misunderstood. Figure 4b 5b, 6b, 7b show the $O^+/H^+$ density ratio used the broad energy range (as mentioned in Response to comment 6). While Figure 4c, 5c, 6c, 7c show the $O^+/H^+$ fluxes ratio at several individual energy ranges. We didn't calculate the $O^+$ or $H^+$ density from one of the energy ranges discussed in Line 202-209. So this relevant part of the description has been amended in my revised manuscript, see Line 277-285 in "Tracked change" manuscript.

**Comments 8:** Line 254-256: After addressing the above comments on how the ion densities were calculated, the authors should briefly address whether these comparisons of density across missions are relevant. For example, if the $O^+$ density (calculated over defined HPCA energy range) is higher than seen by Cluster (calculated in what energy range and using which instrument?), what does this mean? Were the instrument energy ranges equivalent or similar? Otherwise, the direct comparison may not be meaningful.

**Response:** Thanks for the referee's good evaluation and kind suggestion. This comment is very important. From Line 254-256, we can't exclude the reason that Bouhram et al., 2005 used somewhat different energy range for $O^+$ observations result in lower $O^+$ density in their study than mine. The direct comparison can't be meaningful. In this study, the $O^+$ density calculated using HPCA distribution functions at energies from 1 to 40 keV, but Bouhram et al., (2005) used CODIF distribution functions at energies from 3 to 40 keV to avoid contamination from high $H^+$ fluxes. The composition and distribution function (CODIF) analyzer on the Cluster that measures 3-D distributions of the major ion species over the energy range 30–40000 eV. This contrast study is not rigid in this study, so we removed the relevant part in our revised manuscript.

**Comments 9:** Line 305: Since 31 events are not a large number, recommend the authors produce a table to list the dates and times of each of these events so that others in the space science community can also investigate the events for follow-on studies. Such a table could go in an appendix.

**Response:** Yes, this is a good suggestion. I have prepared such a table to list the dates and times of each of these events for follow-on studies in an appendix (see the appendix).

**Comments 10:** All the references in the manuscript need to be checked. For example, all the MMS instrument papers were referenced but do not appear in the references list. It is likely any other references have been missed. It is likely many other references have been missed.

Line 106: Pollock et al. (2016) is referenced but does not appear in the references list
Line 105: Russell et al. (2016) is referenced but does not appear in the references list

Line 104: Ergun et al. (2016) is referenced but does not appear in the references list

Line 104: Lindqvist et al. (2016) is referenced but does not appear in the referenceslist

Line 107: Young et al. (2016) is referenced but does not appear in the references list

**Response:** Thanks for the referee's kind suggestion and carefully evaluating this paper. This mistake should have avoided in the manuscript submission. We added the MMS instrument papers citations in the references list. We also checked carefully all the references in the manuscript to make sure all the citations in the references list. The other spelling and syntax errors have also been checked and corrected in the revised paper.

**Reply to reviewer #2**

Dear reviewer:

We are very grateful to your comments for the manuscript and thanks for carefully evaluating this manuscript. According to your advice, we amended the relevant part of the manuscript. The one-to-one responses to your comments are the following.

**Major comments**

**Comments 1**:One of the conclusions of the manuscript is that particles are transported from the tail towards the dayside. To make such a conclusion more rigid one should show the anisotropy of the particle distributions, which would indicated that particles move from the tail towards the dayside. The oxygen ions could also come from other sources such as inner magnetosphere (filled directly from the nightside aurora into the ring current), from the diamagnetic cavities/cusp (e.g. Slapak et al., Ann. Geophys. 2013,10.5194/angeo-31-1005-2013).

**Response:** Thank you for pointing this out. Yes, the oxygen ions at the dayside LLBL have many sources such as the ring current in the inner magnetosphere, the high latitude auroral region and the cusp. Our paper focuses on the oxygen ions in the duskside magnetopause during intense substorms with AE >500nT. Previous research work has reported that the oxygen ions transferred faster into the ring current in the inner magnetosphere and then they are decayed at the dayside magnetopause under southward IMF or with their large gyroradius effect [e.g., Zong et al., 2001]. Under intense geomagnetic activities such as intense substorms and storms, the oxygen ions from the nightside aurora along the plasma sheet or plasma sheet boundary layer can be fast transferred into the near-Earth magnetotail and then injected into the ring current [e.g., Duan et al.,2017 JGR; Yu and Ridley,2013 JGR]. Recently, Kronberg et al. [2014] reported that the oxygen ions distribution was really anisotropic at the dawn-dusk equator plane. Our observation result is consistent with their report. I have to admit making such a conclusion is not rigid. Because we can't exclude other origins. I removed this expression in my revised paper.

Kronberg, E. A., Ashour-Abdalla, M., Dandouras, I., Delcourt, D. C., Grigorenko, E. E., Kistler, L. M.,…Zelenyi, L. M. (2014). Circulation of heavy ions and their dynamical effects in the magnetosphere: Recent observations and models. Space Science Reviews, 184(1-4), 173–235. https://doi.org/10.1007/s11214-014-0104-0

Yu, Y., and A. J. Ridley (2013), Exploring the influence of ionospheric O+ outflow on magnetospheric dynamics: dependence on the source location, J. Geophys. Res. Space Physics, 118, 1711–1722, doi:10.1029/2012JA018411

Zong, Q.-G., B. Wilken, S. Y. Fu, T. A. Fritz, A. Korth, N. Hasebe, D. J. Williams, and Z.-Y. Pu (2001), Ring current oxygen ions escaping into the magnetosheath, J. Geophys. Res., 106(A11), 25,541–25,556.

**Comments 2:**I am not sure if one could make firm conclusions about dependence on the IMF Bz,if from 31 events only 6 events were observed during northward IMF. On my opinion the statistics are too poor for that.

**Response:** Thank you for the comment. The events of energetic oxygen ions at the duskside magnetopause during intense substorms in our studies are chosen from MMS Phase 1. Because there are limited number of events of intense substorms when MMS passes through the duskside magnetopause during the Phase 1. So the $O^+$ abundance dependence on IMF Bz is not clear. In the revision processes, We added the 26 event satisfied with the criterion into our work. Our work presents 57 intense substorms events with 50 events under the sourthward IMF Bz and only 7 events under the northward IMF Bz. The intense substorms are usually occurring duing southward IMF Bz .This is consistent with the usually external condition of intense substorms [Lyons et al., 2005]. On the other hand, We have added 26 events satisfied with the criterion in the revised paper for better study. We found the $O^+$ density shows expontial decrease with IMF Bz from -10 to 0 nT. This conclusion will be substitute with a more rigid expression in my revised manuscript. -->"When the IMF is southward, the $O^+$ density shows an exponential increase with the IMF Bz absolute value.", as shown in "Tracked change" manuscript Line 31-32.

Lyons, L. R., D.-Y. Lee, C.-P. Wang, and S. B. Mende (2005), Global auroral responses to abrupt solar wind changes: Dynamic pressure, substorm, and null events, J. Geophys. Res., 110, A08208, doi:10.1029/2005JA011089.

**Comments 3:**The "intense substorms" are discussed in this study. Were these substorms associated with magnetic storms? Or these are pure substorm events? What is the reason for choosing intense substorms? Inlcuding other substorms may increase the statistics on the IMF dependence.

**Response:**Thanks for the referee's kind suggestion. In this statistical study, 31 magnetopause crossing events during intense substorm (AE>500 nT) were selected. Among them, there are 4 events during the non-storm time (Dst> -25 nT) and 27 events during the storm time (Dst< -25 nT). There are three resons that we focused on investigating the characteristics of energetic oxygen ions at the duskside magnetopause during intense substorms. Firstly, previous studies have reported that the density and energy flux of oxygen ions in the magnetosphere both increased during magnetic activities, such as intense substorm and storms (e.g.,Daglis et al.,1994; Kronberg et al.,2014). During During disturbed times, oxygen ions can be energized due to duskward drift along the dawn- dusk eletric field in the course of their covection from the distant tail towards the Earth. And the oxygen duskward asymmetry is observed at the near-Earth nightside (eg. Nosé et al., 2000; Luo et al., 2014). So l want to realize the relations between magnetail processes with the Oxygen abundance in the duskside magnetpause. Second, Oxygen ions play a significant role in the energy and mass transport in the coupling process of the solar wind-magnetosphere-ionosphere during intense substorms. The responses of energetic oxygen ions at the duskside magnetopause boundary layer to the solar wind conditions during intense substorms have seldom be reported till now. Third, MMS project can provide a good chance to investigate the features of energetic oxygen ions in the dayside magnetopause boundary layer. The previous spacecraft observations provided significant results of oxygen ions mainly focusing on the tail plasma sheet or middle and high latitude region, such as Cluster [e.g., Nilsson et al.,2006;Slapak et al.,2011]. Thus, our investigation can provide new results in the duskside magnetopause.

Daglis, I. A., Livi, S., Sarris, E. T., & Wilken, B. (1994). Energy density of ionospheric and solar wind origin ions in the near-Earth magnetotail during substorms. Journal of Geophysical Research, 99(A4), 5691–5703. https://doi.org/10.1029/93JA02772

Kronberg, E. A., Ashour-Abdalla, M., Dandouras, I., Delcourt, D. C., Grigorenko, E. E., Kistler, L. M.,…Zelenyi, L. M. (2014). Circulation of heavy ions and their dynamical effects in the magnetosphere: Recent observations and models. Space Science Reviews, 184(1-4), 173–235, doi:10.1007/s11214-014-0104-0.

Ono, Y., M. Nosé, S. P. Christon, and A. T. Y. Lui (2009), The role of magnetic field fluctuations in nonadiabatic acceleration of ions during dipolarization, J. Geophys. Res., 114, A05209, doi:10.1029/2008JA013918.

Luo, H., E. A. Kronberg, E. E. Grigorenko, M. Franz, P.W. Daly, G. X. Chen, A. M. Du, L. M. Kistler, and Y.Wei (2014), Evidence of strong energetic ion acceleration in the near-Earth magnetotail, Geophys. Res. Lett., 41, 3724–3730, doi:10.1002/2014GL060252.

Nilsson, H., et al. (2006), Characteristics of high altitude oxygen ion energization and outflow as observed by Cluster: A statistical study, Ann. Geophys., 24, 1099–1112.

Slapak, R., Nilsson, H., Waara, M., André, M., Stenberg, G., and Barghouthi, I. A.( 2011), O+ heating associated with strong wave activity in the high altitude cusp and mantle, Ann. Geophys., 29, 931–944, doi:10.5194/angeo-29-931-2011

**Comments 4:** Introduction, first two paragraphs can be merged as they contain repeating information about acceleration during dipolarizations. The second paragraph is not completely logical. It would make more sense to describe acceleration of $O^+$ starting from the polar region, then lobe, dipolarizations and then discuss drift. The sentence in lines 43-45 discussing acceleration of electrons during dipolarizations is not really needed as there is a number of references about acceleration of oxygen during dipolarizations in lines 29-47 and the whole text is about $O^+$.

**Response:** Thanks for the referee's kind advice. As you suggested, I adjusted the description order in seconde paragraph to make the introduction more logical and concise.The part of revision can be found in Line 40-65 in "Tracked change" manuscript.

**Comments 5:** lines 90-91, "At present, $O^+$ near the dayside low-latitude magnetopause during substorm expansion phase and recovery phase are still not understood" –> What exactly do you mean under not understood? Which scientific questions are still open? Which questions do you try to answer?

**Response:** Thank you for these comments. Actually, what we want to know is how the $O^+$ abundance ($O^+/H^+$) in the duskside magnetopause varies on AE index and solar wind conditions (e.g. IMF By, IMF Bz, and solar wind dynamic pressure) during the intense substorm (AE >500 nT).The relevant description is revised in "Tracked change" manuscript Line 114-117.

**Comments 6:** lines 91-93, there is paper by Luo et al., JGR, 2017, 10.1002/2016JA023471, in which the energization of $O^+$ at the dayside is discussed. The study also discusses asymmetries of the energetic oxygen due to IMF By and Bz directions. Both IMF By and Bz influence the oxygen abundance at higher energies. However, this is large statistical study and not only cases for the intense substorms. This can be discussed.

**Response:** That would be great. We discussed Luo et al., (2017) results in my revised manuscript. The relevant part can be found in "Tracked change" manuscript Line 399-402. Recently, Using energetic ion composition data at the low latitude dayside magnetopause measured by Magnetospheric Multiscale (MMS) satellites, we study the response of $O^+$abundance ($O^+/H^+$) to the both IMF By and Bz and not only cases for the intense substorms. We found that they indeed influence the oxygen abundance even at lower energies (1-40 keV) and more significant duskside asymmetry of $O^+$ under southward IMF with positive IMF By. These results are consistent with those of Luo et al.,(2017).

**Comments 7:**lines 125-126, 130-131, 180-181, please provide a more precise definition of the substorm onset and recovery phase. For example in paper by Newell and Gjerloev,JGR, 2011, 10.1029/2011JA016779, is a nice example on how to define substorm onset, also using more precise SML index available at the SuperMAG. I do not think that definition when "AE index significantly increases" is a precise one. I do not think that one should provide twice the information about substorm onset in lines 125-126 and 130-131. I would remove the second sentence.

**Response:** Thanks for constructive comments and nice recommendation. We have added a more precise definition of the substorm onset, expansion phase and recovery phase in our revised manuscript. The second information about substorm phase description in lines 125-126 and 130-131 has been removed. We added AU, AL index in Figure 1 to help us identify the phase of a substorm. First, we determined the time interval of the magnetopause boundary layer crossings in each event. Then, we find out how the substorm indices change during that interval from the OMNI data. As Figure 1 shown, the time interval of the magnetopause boundary layer crossing is indicated by the two blue dashed lines. As we know, the AE index is defined as AE=AU-AL. Generally,the substorm onset time is characteristic by the AL index starts to significantly decrease and the AE index significantly increase. During the substorm expansion phase, the AL index will decrease significantly. The interval of the AL index decrease from onset to its minimum is defined as the substorm expansion phase. Then it starts to increase and the interval of the AL index increase from the minimum to the quiet time level is regarded as the substorm recovery phase. In our event, the MMS4 crossed the magnetopause boundary layer from 15:25:10 to 15:36:50 UT on 3 October 2015. From Figure 1f, the AL index reached its minimum ~-750 nT and AE index reach the peak ~1000 nT at about 15:20 UT, then it started to increase to ~ -200 nT at the rest time of interest. So the magnetopause boundary layer crossing occurred during the intense substorm recovery phase. (see Line 167-180 in "Tracked change" manuscript)

[Figure]

Figure 1. The three components IMF Bx, By, Bz, solar wind dynamic pressure, as well as AU, AL, and AE index from CDAweb OMNI data.

**Comments 8:** lines 179-180, actual observations of the IMF and solar wind dynamic pressure could be used directly from the MMS observations at the magnetopause crossings. This would be much more precise.

**Response:** Getting the much more precise IMF and solar wind dynamic pressure would be better. When the IMF passes through the bow shock, its direction would be changed in the magnetosheath. We will compare the IMF and solar wind dynamic pressure directly from the MMS-with those from OMNI data in the detailed events analysis. Then we will choose the more precise data.

**Comments 9:** lines 277-278, For higher energies the larger statistics one can clearly see that the stronger duskward asymmetry in the plasma sheet and the dayside magnetosphere is observed under the southward IMF, e.g. Luo et al., JGR, 2017. One should mention that no influence of IMF Bz is observed in case of the energies below 40 keV and for 31 intense substorm events.

**Response:** Thanks for your constructive suggestion.We agree with your comments. Recently, We used energetic ion composition data at the dayside magnetopause measured by Magnetospheric Multiscale (MMS) satellites, we study the response of $O^+$ abundance to IMF Bz. The $O^+$ abundance showing strong duskside asymmetry in the magnetopause boundary layer under southward IMF than that under northward IMF also be found in our study, which is consistent with Luo et al.2017 result. As you suggested, the influence of IMF Bz is not clear in our 31 intense substorm events. This description in Lines 277-278 has been removed.

**Comments 10:** lines 286-287, the energetic $O^+$ occurs predominantly under southward IMF. Here I would say that it was chosen to be like this. Choosing the intense substorms one increases the probability of observing the southward IMF quite significantly. This also contradicts to statement in the lines 277-278, that IMF Bz does not influence abundance of $O^+$ at the magnetopause. There is not enough provided data to conclude so.By increasing the number of events under the northward IMF one may see a different picture. One can see pretty nice trend in Figure 6b, that the abundance is increasing with the decrease of IMF Bz at least for the expansion phase. Generally on my opinion there is not enough statistics in this study to make conclusions about IMF dependence. One should expand the statistics

**Response:** Thanks for your valuable comments. We agree with you that choosing the intense substorms one increases the probability of observing the southward IMF significantly. To make convincing conclusions about IMF dependence, we expanded the statistic to 57 events. Some conclusions may be still not convincing due to not enough statistical events. As the MMS operate longer, more magnetopause crossing during intense substorm will be detected. It will be helpful. The sentences in Line 286-287 have been deleted.

**Comments 11:** lines 304-306, this conclusion is not supported by the observations. Just looking at the scatter points of the number density, I do not see a statistically significant difference between these two phases. One should either show fits to those points or bin them according to some parameters and show that the difference is significant.

**Response:** Thanks for your comments. At the beginning of this study, we focus on the response of $O^+$ abundance on the geomagnetic activity and solar wind conditions during intense substorms. Because the magnetosphere has the different dynamics in the near-Earth space during the different phase of intense substorms, especially in substorm expansion phase and recovery phase. So we want to investigat variations of energetic O$^+$ density at the duskside magnetopause boundary layer duing different phases of intense substorms. Due to the number of events are limited (only 26 events during expansion phase), we don't think it makes sense to fit those points or bin then according to some parameters. As the MMS operating longer, more magnetopause crossing during intense substorm will be detected. It will be helpful. Our selecting events we drawn our summary on the energetic O$^+$ density as description in the last part of our manuscript. In generally, the O$^+$ in the magnetosphere are origin from the ionosphere and transferred into the different magnetosphere region during magnetic activities. A excellent review paper of this aspect has been reported by Keika et al., (2013). Our new results from MMS data provide another support of previous studies.

Keika, K., L. M. Kistler, and P. C. Brandt (2013), Energization of O+ ions in the Earth's inner magnetosphere and the effects on ring current buildup: A review of previous observations and possible mechanisms, J. Geophys. Res. SpacePhysics, 118, 4441–4464, doi:10.1002/jgra.50371.

**Comments 12:** lines 313-315, energetic oxygen ions also indicate the transport at the dayside magnetosphere (e.g. Liao et. al, JGR, 2010, 10.1029/2010JA015613). These different transports are hard to distinguish (e.g. Luo et al., JGR, 2017).

**Response:** Thanks for your comments and paper recommendation. Liao et al., (2010) JGR and Luo et al.,(2017) JGR are both cited in our revised manuscript (see see Line 391-397 and Line 399-402 in "Tracked change" manuscript). I agree with you, these different transports are hard to distinguish. Also, the different transports of oxygen ions from the ionosphere to different part of the magnetosphere are significant and interest. It is outside the focus of our manuscript. We will investigate this issue with conjunction observations by multiple spacecraft in different magnetosphere locations. This conclusion in Line 313-315 has been deleted.

**Comments 13:**Figures 4-7, just looking at the scatter plots it is hard to make certain conclusions.One should either bin the points to show the average trend or fit them with some dependences and increase the number of events.

**Response:** Thanks for your nice suggestions. I have binned the points to show the average trend before submitted this manuscript. As you said,the number of events is too low, so the trend is not obviously or has low credibility and we abandoned this method. As you suggested, we fit the oxygen density dependence on the IMF Bz and Psw, respectively. Recently, Using energetic ion composition data at the low latitude dayside magnetopause measured by Magnetospheric Multiscale (MMS) satellites, we study the response of H$^+$, O$^+$density and their ratio to the geomagnetic activity (indicated by SYM-H index) and solar wind conditions (including interplanetary magnetic field (IMF) By, IMF Bz and solar wind dynamic pressure). In this study, we bin the points due to enough events. Our new manuscript has been submitted to the JGR.

**Minor comments:**

1. **Line 19:** What is the energy range of the oxygen observations used in this study? Please indicate the upper energy limit in the abstract. This is important to know when assessing the number densities.

**Response:** In this study, only the $O^+$ at energies from 1 keV to 40 keV measured by HPCA are used. The upper energy limit of HPCA is 40 keV. This information is added to the abstract. (see Line 18 in "Tracked change" manuscript )

2. **Line 45:** I did not find the reference to Lui et al., 1999 in the reference list. (see Line 589-591 in "Tracked change" manuscript )

3. **Line 47:** "during activity geomagnetic disturbance" –> "during disturbed geomagnetic activity" (see Line 50-51 in "Tracked change" manuscript )

4. **Line 55:** "[e.g. Yau and Andre, 1997]. And then..." –> "[e.g. Yau and Andre, 1997] Then..." (see Line 62 in "Tracked change" manuscript )

5. **Line 85:** please remove one "However". (see Line 109 in "Tracked change" manuscript )

**Response to comments from 2-5:** The above expression errors have been checked and corrected. The missing reference has been added to the revised manuscript.

6. **Lines 106-107**: Does HPCA distinguish between $O^+$, $N^+$, and $C^+$? Or what measures actually the CNO group?

**Response:** "the HPCA is a time-of-flight (TOF) mass spectrometer designed to measure the velocity distributions of the four ion species ($H^+$, $He^{++}$, $He^+$ and $O^+$ ) known to be important in the reconnection process. The measurement technique is based on a combination of electrostatic energy-angle analysis with time-of-flight velocity analysis. The result is an accurate determination of the velocity distributions of the individual ion species. In order to meet the stringent scientific requirements of the MMS mission, the HPCA incorporates three new technologies. The first extends counting rate dynamic range by employing a novel radio frequency mass filter that allows minor species such as $He^{++}$ and $O^+$ to be measured accurately in the presence of intense proton fluxes found in the dayside magnetopause. The second ensures that TOF processing rates are high enough to overlap with the low end of the RF dynamic range, while the third enhances ion mass resolution.

[Figure]

**Fig. 20** TOF spectrum for four ion species and background ($H_2^+$ is a substitute for $He^{++}$ and $N^+$ is a substitute for $O^+$). *Red areas* demarcate bins that define ion species and background. The *peak* at ~200 ns corresponds to $N_2^+$

**Fig. 31** TOF boundaries as a function of energy for four ion species plus background

During each energy scan a data set consisting of 63 TOF spectra × 512 TOF bins × 16 elevations is accumulated and histogrammed. The resulting TOF spectra are then parsed into five bins that define the ion species H+, He++, He+, O+ and background (Fig. 20). The red portions of the spectrum in Fig. 20 indicate typical species boundaries. Since ion times-of flight are both mass and energy-dependent the range of TOF limits for each species changes with energy (Fig. 31)". (the Figure and description are cited from Young, D. T., Burch, J. L., Gomez, R. G., De Los Santos, A., Miller, G. P., Wilson, P., et al. (2016). Hot Plasma Composition Analyzer for the Magnetospheric Multiscale Mission. Space Science Reviews, 199(1–4), 407–470, doi:10.1007/s11214-014-0119-6.).

As for this interesting question, I specially contacted the HPAC PI (Stephen Fuselier), he replied me **"I'm working right now to see if we can see C+ and possibly N+ in the mass spectra. They would not appear as a separate mass peak because of straggling in the foil. I'm not sure if we can even tell if they are there. What we bring to the ground and call O+ could contain substantial N+. The C+ peak would probably be at a lower time-of-flight than what we bring to the ground, but you could safely say that what we call O+ could be N+O+."**

7. **Line 122:** "At the beginning of the time interval, the solar wind dynamic pressure..."–>The dynamic pressure is only at the begging of the time interval about 2 nPa. (see Line 161-162 in "Tracked change" manuscript )

8. **Lines 124-125**: I would change to " These solar wind conditions led to the intense substorm (AE>500 nT). (see Line 166 in "Tracked change" manuscript )

9. **Lines 148-150:** Figure 2, I would say that the fluxes at energies below 2 keV in Figure 2j is also contamination. This should be mentioned also in Figure caption and even better when it is indicated on the plot itself. (see Figure 2j red box in revised manuscript )

**Response to comments from 7-9:** Thanks for the referee's valuable suggestion. The minor comments 7 and 8 have been corrected in my manuscript. The red box indicating the $O^+$ contamination from high proton fluxes was plotted in figure 2j and relevant description was mentioned in the Figure 2 caption (see Line 676-677).

10. **Line 195:**I would remove "On the other hand".

**Response:** This is a common colloquial expression. We removed it.

11. **Lines 221-224:** These results also agree with Kronberg et al., JGR, 2012,10.1029/2012JA018071 which showed for 10 keV $O^+$ strong increasing under the duskward IMF indicated by the clock angle in the inner magnetosphere.

**Response:** That would be great. We cited this paper in the relevant part of the revised manuscript to increase valid of our results. see Line 387-389 in "Tracked change" manuscript )

12. **Lines 251-268:** another reason can be that *Bouhram et al.,* 2005 have used somewhat different energy ranges for $O^+$ observations.

**Response:** Yes, We agree with you. I can't exclude the reason that *Bouhram et al.,* 2005 used somewhat different energy range for $O^+$ observations. In this study, the $O^+$ density calculated using HPCA distribution functions at energies from 1 to 40 keV, but Bouhram et al., (2005) used CODIF distribution functions at energies from 3 to 40 keV to contamination from high $H^+$ fluxes. This contrast study is not rigid in this study. We removed the relevant part in my revisited manuscript.

13. **Line 276:**magenetopause –> magnetopause.(see Line 384 in "Tracked change" manuscript )

14. **Line 279:** have –> has. (see Line 390 in "Tracked change" manuscript )

15.**Line 287:** dominated occurring –> occurs predominantly.(removed)

17. **Lines 296-297:** I would change this sentence to "The reconnection rate is likely will be reduced by the mass-loading but not suppressed at the magnetopause [*Fuselier et al.,* 2019]. (see Line 429-430 in "Tracked change" manuscript )

17. **Figure 1**, caption, "The three components of the IMF, Bx, By, Bz..." (see Line 661 in "Tracked change" manuscript )

18. **Figure 2,** I would indicate on the plot contamination. In the caption, line 481 (k)–>(l). (see Line 671 in "Tracked change" manuscript )

**Response to comments from 13-18**: Thanks for referee'scarefully evaluating this paper and important suggestions. We have revised the above errors and plotted the red box indicating the $O^+$ contamination from high proton fluxes in figure 2j. The other spelling and syntax errors have been checked and corrected. We acknowledge the reviewer's comments and suggestions very much, which are valuable in improving the quality of our manuscript.

**Reply to reviewer #3**

Dear reviewer:

We are very grateful to your comments for the manuscript and thanks for carefully evaluating our manuscript. We acknowledge the reviewer's comments and suggestions very much, which are valuable in improving the quality of our manuscript. According to your advice, we amended the relevant part of the manuscript. Responses to your comments are below point by point.

**Comments 1**: Lines 90-95: There is a lot of information leading up to this point in the introduction, however with the lines preceding and in this paragraph itself, it is unclear what is not well understood and how/what this paper will provide to answers to. Currently, the introduction reads as a quite thorough list of previous studies, but it is not readily apparent how they string together, and what they are necessarily building up to. I would suggest stating what the paper will study before this point and tailoring the introduction to build off of that somewhat, because at this point as a reader it is still unclear.

**Response:** Thanks for the referee's kind advice. As you suggested, we did some revisions in our revised manuscript. We adjusted the first two paragraphs to make the introduction more logical and concise (see Line 40-65 in "Tracked change" manuscript). The introduction is organized as following orders. First, we stress the importance of $O^+$ during the intense substorms, describe acceleration of $O^+$ starting from the polar region, then lobe, near-Earth plasma sheet and then discuss drift. Second, we describe the $O^+$ behavior in the magnetopause. Third, we referred to the $O^+$ density dawn-dusk asymmetry in the magnetopause. Finally, we describe the questions what this paper tries to answer. As the following described: "At present, variations of $O^+$ abundance ($O^+/H^+$) in the dusk flank magnetopause during intense substorms (AE >500 nT) on AE index and solar wind conditions (e.g. IMF By, IMF Bz, and solar wind dynamic pressure) are still not understood. Previous studies of $O^+$ during intense substorms mainly focused on $O^+$ energizations in the NEPS in the magnetotail (e.g., Duan et al., 2017; Nosé et al., 2000; Ohtani et al., 2011). At present, The Magnetospheric Multiscale (MMS) mission gives us an opportunity to focus on the $O^+$ in the low latitude dayside magnetopause region. In this study, we mainly investigate statistical features of energetic $O^+$ in the dusk flank magnetopause and their relations with AE index and solar wind conditions (e.g. IMF By, IMF Bz and solar wind dynamic pressure) during the intense substorms. (see Line 114-124 in "Tracked change" manuscript)

**Comments 2:** Lines 128-150: HPCA & FPI fluxes are in differential flux and energy flux units. Is there a benefit in having their fluxes in different units? If they are to remain, a point should be included in the text that the units are different.

**Response:** Thanks for the referee's kind suggestion. We described the HPCA and FPI fluxes having different units in our revised manuscript. Figures 2g and 2h show the electron omnidirectional differential energy fluxes and ion omnidirectional differential energy fluxes, respectively. Figure 2i to 2l presents the differential particle fluxes of $H^+$, $O^+$, $He^+$, $He^{++}$, respectively. To better identify the fluxes variations at specific energies, we choose the ion and electron fluxes from FPI in the energy flux unit. The relevant description has been add into the Line 195-198 in "Tracked change" manuscript)

[Figure]

Figure 2. The energetic $O^+$ is observed at the magnetopause during an intense substorm on 03 October 2015 by MMS 4. From top to bottom are (a) the magnetic field three components, Bx (blue line), By (gree line), Bz (red line) and the total magnitude Bt (black line), (b) the electric field three components, Ex (blue), Ey (gree) and Ez (red), (c) Ion parallel (red) and perpendicular(black) temperatures, (d) The density of ion (green) and electron (blue), (e) three components of the ion velocity, (f) the $H^+$(overthe full HPCA energy range) and $O^+$ (at energies from 1 to 40 keV)densities, (g) electron omnidirectional differential energy fluxes, (h) ion omnidirectional differential energy fluxes,(i) to (l) present differential particle fluxes of $H^+$, $O^+$, $He^+$, $He^{++}$, respectively. The Geocentric Solar Magnetospheric (GSM) coordinate system is adopted. The thick bars at the top of the panel present different regions encountered on this magnetopause crossing event. The orange and blue bars represent the magnetosheath and the magnetosphere, respectively. The green bar represents the magnetopause boundary layer. The black horizontal line in figure 2j is at 1keV and the $O^+$ contamination from high $H^+$ fluxes is indicated by the red box. The FPI data in Figure 2(c-e) and (g-h) are from FPI L2 data products and in the fast mode.

**Comments 3:** Lines 128-150: The HPCA flux in panels i-l have artificial striping every 4 energy bins due to way HPCA determines the count rate over 4 energy channels in survey mode. It would be best to correct this, however, describing the artificial striping would also be sufficient. I am also not certain that these HPCA fluxes are actually omni-directional as they do not appear to be half-spin averaged, please verify.

**Response:** Thanks for your important comments. "The HPCA flux in panels i-l have artificial striping every 4 energy bins due to way HPCA determines the count rate over 4 energy channels in survey mode." The above sentence has been added into the Line 194-195 in "Tracked change" manuscript. Figure 2i to 2l presents the differential particle fluxes of $H^+$, $O^+$, $He^+$, $He^{++}$, respectively. They are actually not Omni-directional and not half-spin averaged. We corrected this description in our revised manuscript. These differential particle fluxes of $H^+$, $O^+$, $He^+$, $He^{++}$ calculations are used The Space Physics Environment Data Analysis System (SPEDAS) software package. More details about SPEDAS can be found in Angelopoulos et al. (2019) and cited as (Angelopoulos, V., Cruce, P., Drozdov, A. et al. Space Sci Rev (2019) 215: 9. https://doi.org/10.1007/s11214-018-0576-4). We also cited this paper in our revised manuscript (see Line 478-479 in "Tracked change" manuscript).

**Comments 4:** Lines 134-137: Please describe where the FPI/HPCA moments shown come from. This is quite important since the majority of the results presented are dependent on these moments.

**Response:** Thanks for the referee's kind advice. We have clarified where the FPI/HPCA moments shown come from. We have added detailed information about moments into Line 198-207 in "Tracked change" manuscript. The plasma moments (e.g. Ion parallel and perpendicular temperatures, ion, and electron densities and ion velocity) from FPI shown in Figure 2c-e are all from MMS L2 data products. They are default moments calculated over the full FPI energy range from 10 eV to 30 keV. But the $O^+$ density shown in Figure 2f is recalculated from HPCA distribution functions in the range of energies from 1 to 40 keV. From the $O^+$ fluxes shown in Figure 2j, there still exist a large number of fluxes below 1 keV in the magnetosheath. This part of $O^+$ fluxes is fake and contamination from high proton fluxes. So we consider the number density of $O^+$ with energies from 1 to 40 keV. It is more appropriate to represent the true $O^+$ in the magnetopause. While the $H^+$ density which computed over the full HPCA energy range from 1eV to 40 keV from L2 data products are used in Figure 2f.

**Comments 5:** Figures 1-2: I would suggest using these two figures to establish the criteria for the statistical study. In my opinion, more text should be added that describes a greater context for these 2

figures inclusion. Either establishing points that lend themselves to the paper's conclusion and/or use the figure to establish conditions for the statistical study.

**Response:** Thanks for your nice comments. In this statistical study, First, we identified the magnetopause crossing event (complete magnetopause crossing from the magnetosheath to the magnetosphere, vice versa) during phase 1 from the summary plot in https://lasp.colorado.edu/mms/sdc/public/plots/. Then we plotted the more detailed overview of these events to identify the magnetopause boundary layers, as Figure 2 shown. Figure 2 was mainly used to determine the magnetopause boundary layer crossing interval. Only events that AE index larger than 500 nT during the magnetopause boundary layer crossings interval were selected. Finally, the mean value of the $H^+$, $O^+$ density and their fluxes shown in Figure 2 were calculated in the magnetopause boundary layer. Correspondingly, the AE index, IMF By, Bz and solar wind dynamic pressure from the OMNI data system shown in Figure 1 were also averaged during that interval. Figure 1 mainly provided the corresponding solar wind conditions and AE index. The above expressions have been added into Line 245-256 in the "Tracked change" manuscript.

**Comments 6**: Lines 176-181: This is one of the more major comments on the paper. The current description of the event selection criteria is not sufficient. Interpretation of a statistical study is almost entirely dependent on understanding how the statistical study is conducted. It is currently not clear what the criteria for event selection is. Is it any MP crossing with AE > 500? Why was 500 chosen as a threshold in AE (i.e. stats are somewhat low, would AE > 300 or 400 provide more events and still be "intense"?).

**Response:** Thanks for your valuable comment. The magnetopause crossing event in our statistical study all during the intense substorm (AE > 500nT). How we chose these events is replied in the before comment. The reason why we choose intense substorm with AE >500 nT is based on the results from Daglis et al (1994) (Figure 6 in this reference, as shown in the below). They found that the $O^+$ energy density has a great correlation with the AE index in the near-Earth plasma sheet (NEPS). During the intense substorm expansion phase, $O^+$ energy density explosively increases with AE index in the range of larger than 500nT. Otherwise, Lennartsson and Shelley, (1986) pointed out that the ion composition had a large variance at substorm. During the intense disturbed conditions (AE~1000nT), the increase in the O+ energy density is strongest around local midnight where $O^+$ become the most abundant ion. The previous researches of oxygen ions during intense substorms are mainly focused on the nightside NEPS. Thus, we want to know whether the $O^+$ abundance in the dusk flank magnetopause varies on AE index and solar wind conditions during the intense substorm and how it changes with the above parameters. Characteristics of Oxygen ions in the high latitude polar region and near-Earth magnetosphere during intense magnetic activities have been investigated deeply and widely. But O$^+$ abundance in the dayside magnetopause has seldom been reported during intense substorms.

[Figure]

Ion energy density at substorm expansion-phase
AMPTE/CCE - CHEM spectrometer (58 events)
Exponential fit: b=2.23x10$^{-3}$, a=0.16, r=0.37, F=9.12

**Figure 6.** Same format as Figure 2; all substorms with $AE_{exp} \geq 500$ nT (encircled symbols) are excluded from the fit.

**Comments 7:** How exactly is the magnetopause boundary layer determined? Is there any consideration for if the substorm is during a storm or the 1st/2nd/3rd in a series of substorms? Specifically, how are substorm phases determined? What is meant by the mean value of the flux (over a range of energies, one energy)? How long were the average events? Please provide greater context for the choices of criteria used in this study.

**Response:** The magnetopause boundary layers are identified here primarily through plasma fluxes and moments. The magnetopause boundary layer can have densities and temperatures between that of the magnetosphere and the magnetosheath. Meanwhile, the magnetopause boundary layer shows the gradient of the energy flux of particles and number density and magnetic field obvious. Ion jets are also signatures of passing through the magnetopause boundary layers. In this study, the separatrix between the magnetosheath and the magnetopause boundary layer is determined by the appearance of the magnetospheric electron. Similarily, the separatrix of the magnetosphere and the magnetopause boundary layer is determined by the magnetosheath electron disappearance. The mean value of the H$^+$, O$^+$ density and their fluxes are calculated in the magnetopause boundary layer. Correspondingly, the AE index, IMF By, Bz and solar wind dynamic pressure from the OMNI data system were averaged during the time interval of magnetopause boundary layer crossing. As Figure 2 shown, the time interval of the magnetopause boundary layer crossing is marked by the two blue dashed lines. As we know, the AE index is defined as AE=AU-AL. Generally, the substorm onset time is characteristic by the AL index starts to significantly decrease and the AE index significantly increase. During the substorm expansion phase, the AL index will decrease significantly. The interval of the AL index decrease from onset to its minimum is defined as the substorm expansion phase. Then it starts to increase and the interval of the AL index increase from the minimum to the quiet time level is regarded as the substorm recovery phase. In our event, the MMS4 crossed the magnetopause boundary layer from 15:25:10 to 15:36:50 UT on 3 October 2015. From Figure 1f, the AL index reached its minimum ~-750 nT and AE index reach the peak ~1000 nT at about 15:20 UT, then it started to increase to ~ -200 nT at the rest time of interest. The two blue dashed lines indicate the time interval of the magnetopause boundary layer crossing. According to the variation and peak value of the AU, AL and AE index in Figure 1e to 1g.The magnetopause boundary layer crossing occurred during the recovery phase of this intense substorm. The mean value of the flux is over two energy ranges close to the typical energy such as 1 keV, 10 keV and etc. We didn't consider if the substorm in during a storm or the 1st/2nd/3rd in a series of substorms. In this statistical study, 31 magnetopause crossing events during intense substorm (AE>500 nT) were selected. Among them, there are 4 events during the non-storm time (Dst> -25 nT) and 27 events during the storm time (Dst< -25 nT).These detailed contexts for choices of criteria used in this study are described Line 246-250 in the "Tracked change" manuscript.

[Figure]

Figure 1. The three components of the IMF Bx, By, Bz, solar wind dynamic pressure, as well as AU, AL, and AE index from CDAweb OMNI data.

**Comments 8:** Lines 179-180: One of the main points from this paper is that the high-density $O^+$ can be transported from the nightside tail to the magnetopause where it is observed. Please discuss any effect (or lack there of) of using OMNI solar wind values at the bow shock to correlate with observations of high $O^+$ density which is being driven by processes which invariably take some amount of time to occur.

**Response:** Thanks for the referee's good evaluation and kind suggestion. Making such a conclusion is not rigid. I didn't give direct evidence to prove that these $O^+$ are transported from the tail towards the dayside. So, I deleted this expression in my revised paper.

**Comments 9:** Lines 203-205: With the decimation of HPCA fluxes during survey mode, the count rate is recorded/distributed over 3-4 energy channels. With this in mind, is it appropriate to describe the comparisons of the flux as being over such a small energy range, since the flux/count rate could have been dominated by a nearby energy channel? Potentially, it would be more accurate to re-bin the HPCA flux into 16 energy channels instead of 63, and compare the >1 keV flux levels of these larger energy bins. Please discuss, currently it seems a bit misleading to describe the flux as being over such a narrow energy range.

**Response:** Thanks for the referee's nice comment and kind suggestion. The main purpose of calculating the $O^+/H^+$ particle fluxes ratio is to study the $O^+$ abundance at different energies on AE index and solar wind conditions (e.g. IMF By, IMF Bz and solar wind dynamic pressure) during the intense substorms. Since the energy range of $O^+$ and $H^+$ in the HPCA are the same. So we directly divide $O^+$ particle fluxes by H+ particle fluxes and we mainly focus on the ratios and not the values of their fluxes at specific energies.

Comments 10: Lines 231-236: Here it is stated that, "the maximum number density of energetic O+ at the dusk flank magnetopause is during the intense substorms recovery phase under the southward IMF. But the maximum ratio of $n(O^+)/n(H^+)$ at the dusk flank magnetopause is during intense substorm recovery phase under the northward IMF. IMF Bz seems play a minor role in O+ abundance at the dusk flank magnetopause during intense substorm." It is not clear from the data as it is presented that this is true. The density ratio is of course dependent on O+ and H+ (which can come from the ionosphere and the solar wind). Comparing Figures 4a and 5a, it is not clear to me by eye that n(O+) is more dependent on By than Bz. It very well may be, but it is not readily apparent. Thus, is the density ratio difference actually from O+ or H+? Additionally, only 6 of the events in the study have a Bz > 0. This is notable, as Bz not being random does have an impact on the events. Thus, from this study it appears that Bz does play a role in the events being studied.

**Response:** Thanks for your valuable comments. Some descriptions determined by eye are not convincing. So I add the detailed information about density and corresponding IMF conditions in the supplement materials. The sentences in Line 231-136 have been revised by "The maximum density of energetic $O^+$ at the dusk flank magnetopause is under the southward IMF. Meanwhile, the maximum $O^+/H^+$ density ratio at the dusk flank magnetopause is under the southward IMF." The conclusion of "IMF Bz seems to play a minor role in $O^+$ abundance at the dusk flank magnetopause during intense substorm." In this manuscript is not rigid. So the relevant description has been removed. It noted that choosing the intense substorms one increase the probability of observing the southward IMF significantly. We found a nice trend that O$^+$ abundance increase with the IMF By. From Figure 6b, the O$^+$/H$^+$ density ratio show an obvious decrease with IMF Bz from -2 to 2 nT during the recovery phase (red crosses shown). Due to not enough statistical events (only 6 of the events in the study with northward IMF), some conclusions may be not convincing. As the MMS operating longer, more magnetopause crossing during intense substorm will be detected. It will be helpful. The relevant part "the IMF Bz seems play a minor role in O$^+$ abundance at the dusk flank magnetopause during intense substorm." has been deleted.

**Comments 11:** Lines 241-242: "number density ratio at the dusk flank magnetopause during intense substorms have a weak correlation with the solar wind dynamic pressure." Can you quantify this correlation? In general, there are a lot of points currently that are driven from visual inspection of very scattered plots, when greater statistical rigor perhaps could be applied.

**Response:** Thanks for your suggestions. I agree with you, we need quantify this correlation and by eye is not rigid. So we fit the oxygen density dependence on pressure. Due to the number of events are limited (only 9 events during expansion phase) and distribution plot is very scattered, we don't think it makes sense to bin them according to some parameters. The sentence "number density ratio at the dusk flank magnetopause during intense substorms have a weak correlation with the solar wind dynamic pressure." has been removed. And substituted by more detailed description →"Figure 7a present that the O$^+$ density at the dusk flank magnetopause during intense substorms has a positive correlation with the solar wind dynamic pressure. The empirical functional relation between the O$^+$ density and solar wind dynamic pressure (from 1 to 4.5 nPa) is also established in the Eq.(3) and the corresponding correlation coefficient is 94%. From Figure 7b, the O$^+$/H$^+$ density ratio during recovery phase show a decrease from about 2.5 to 3 nPa. It is also noted that the O$^+$/H$^+$ density ratio increase with solar wind dynamic pressure from ~3 to 4 nPa."

**Comments 12:** Figures 4-7: The captions of the figures mention that the 95% confidence intervals are shown. Please mention this in the text and describe how it is calculated.

**Response:** Thanks for your kind suggestion. In the revised manuscript, we add the error bars in each point indicating a 90% Confidence Interval (CI). How to calculate the CI is described as follows: Step 1: find the number of observations $n$ in the magnetopause boundary layer. Then calculate their mean $\bar{x}$ and standard deviation $s$. Step 2: Find the $k$ value for 90% CI (the $k$ value is 1.65). Step 3: use that $k$ in this formula for the CI:

$$\bar{x} - k\frac{s}{\sqrt{n}} < \mu < \bar{x} + k\frac{s}{\sqrt{n}}$$

Where $\bar{x}$, $s$ and $n$ are the mean value, standard deviation and the sampling number of observations, respectively. $k$ in the above formula can be determined by calculating a 90% CI for each events. See Line 286-292 in the "Tracked change" manuscript.

**Very minor comments:**

1. **Lines 103-106:** Please explicitly state that FPI does not discriminate between different ion species.

   **Response:** thanks for your kind suggestion. We added the "FPI does not discriminate between different ion species" in the Line 136-137 in the "Tracked change" manuscript.

2. **Line 107:** Strictly speaking, HPCA measures up to 40 keV/q (thus for He$^{++}$ this gets up towards 80 keV).

   **Response:** Thanks for you carefully evaluate this manuscript. We agree with you, the HPCA maximum measurement for energy per charge is 40 keV/q. Line 138 in the "Tracked change" manuscript has been revised as you suggested.

3. **Line 116:** The authors might as well finish this thought, that this is due to spacecraft separation/scales of particle motion.

   **Response:** Thanks for your nice suggestion. We added this sentence "this is due to spacecraft separation/scales of particle motion." into Line 150-151 in the "Tracked change" manuscript for finishing this thought.

4. **Line 296:** Fuselise et al. should be Fuselier.

   **Response:** Thanks for you carefully evaluating this manuscript and giving important suggestions. We have revised this error. The other spelling and syntax errors have also been checked and corrected. See Line 430 in the "Tracked change" manuscript.

5. **Lines 304-306:** I would re-phrase this sentence. It is a minor distinction, but it currently reads as if you have studied energetic O+ across the entire magnetopause during substroms and found that the most prevalent region of O+ is the dusk flank during the recovery phase. Whereas, it should be more like, "Observations of energetic O+ at the dusk flank magnetopause during substorms are mainly found within the recovery phase."

   **Response:** Thanks for referee's nice suggestion. We expand the statistical and found 
[revised manuscript text omitted]